# Function and regulation of a steroidogenic CYP450 enzyme in the mitochondrion of *Toxoplasma gondii*

**Beejan Asady**[1], **Vera Sampels**[1], **Julia D. Romano**[1], **Jelena Levitskaya**[1], **Bao Lige**[1], **Pratik Khare**[2], **Anne Le**[2,3¤], **Isabelle Coppens**[1] *

**1** Department of Molecular Microbiology and Immunology, Johns Hopkins University Bloomberg School of Public Health, Baltimore, Maryland, United States of America, **2** Department of Chemical and Biomolecular Engineering, Johns Hopkins University, Baltimore, Maryland, United States of America, **3** Department of Pathology and Oncology, Johns Hopkins University School of Medicine, Baltimore, Maryland, United States of America

¤ Current address: Gigantest, Inc., Baltimore, Maryland, United States of America.
* icoppens@jhsph.edu

**Data Availability Statement:** All relevant data are within the paper and its Supporting Information files.

## Abstract

As an obligate intracellular parasite, *Toxoplasma gondii* must import essential nutrients from the host cell into the parasitophorous vacuole. We previously reported that the parasite scavenges cholesterol from host endocytic organelles for incorporation into membranes and storage as cholesteryl esters in lipid droplets. In this study, we have investigated whether *Toxoplasma* utilizes cholesterol as a precursor for the synthesis of metabolites, such as steroids. In mammalian cells, steroidogenesis occurs in mitochondria and involves membrane-bound type I cytochrome P450 oxidases that are activated through interaction with heme-binding proteins containing a cytochrome b5 domain, such as members of the membrane-associated progesterone receptor (MAPR) family. Our LC-MS targeted lipidomics detect selective classes of hormone steroids in *Toxoplasma*, with a predominance for anti-inflammatory hydroxypregnenolone species, deoxycorticosterone and dehydroepiandrosterone. The genome of *Toxoplasma* contains homologs encoding a single type I CYP450 enzyme (we named TgCYP450mt) and a single MAPR (we named TgMAPR). We showed that TgMAPR is a hemoprotein with conserved residues in a heme-binding cytochrome b5 domain. Both TgCYP450 and TgMAPR localize to the mitochondrion and show interactions in *in situ* proximity ligation assays. Genetic ablation of *cyp450mt* is not tolerated by *Toxoplasma*; we therefore engineered a conditional knockout strain and showed that iΔTgCYP450mt parasites exhibit growth impairment in cultured cells. Parasite strains deficient for *mapr* could be generated; however, ΔTgMAPR parasites suffer from poor global fitness, loss of plasma membrane integrity, aberrant mitochondrial cristae, and an abnormally long S-phase in their cell cycle. Compared to wild-type parasites, iΔTgCYP450mt and ΔTgMAPR lost virulence in mice and metabolomics studies reveal that both mutants have reduced levels of steroids. These observations point to a steroidogenic pathway operational in the mitochondrion of a protozoan that involves an evolutionary conserved TgCYP450mt enzyme and its binding partner TgMAPR.

**Funding:** This work is supported by NIH GRANT NUMBER R01 AI138714 to IC. The funders had no role in study design, data collection and analysis, decision to publish, or preparation of the manuscript.

**Competing interests:** The authors have declared that no competing interests exist.

## Author summary

In addition to controlling membrane fluidity, cholesterol serves as a substrate for the synthesis of important biomolecules, in particular steroid hormones. Steroidogenesis involves mitochondrial cholesterol-metabolizing cytochrome P450s activated by membrane-associated progesterone receptor (MAPR) hemoproteins. The intravacuolar parasite *Toxoplasma* scavenges cholesterol from the host and our targeted lipidomics reveal the presence of pregnenolone and selected steroid hormones in the parasite. We investigated a potential steroidogenic pathway in *Toxoplasma* by characterizing the single CYP450 (TgCYP450mt) and the MAPR homolog (TgMAPR) in *Toxoplama*. Both TgCYP450mt and TgMAPR are expressed in the mitochondrion and interact with each other. Parasites lacking *mapr* are viable *in vitro* unlike ΔTgCYP450mt parasites. Conditional TgCYP450mt knock-out parasites and ΔTgMAPR suffer from developmental defects, are poorly virulent and have reduced levels of steroids compared to WT parasites. These data suggest that *Toxoplasma* has the ability to synthesize some steroids in the mitochondrion, using a cholesterol-metabolizing cytochrome P450 and reveal that steroidogenesis is an ancient pathway, conserved in eukaryotic lineages.

## Introduction

*Toxoplasma gondii* is an obligate intracellular parasite that multiplies in mammalian cells within a self-made membrane-bound compartment, the parasitophorous vacuole (PV) that protects against host cytosolic destructive pathways. Inherent in the adaptation of intracellular microbes to the nutrient-filled host cell interior, *T. gondii* lost many genes involved in the *de novo* syntheses of essential metabolites and thus must salvage and store nutrients from its host to survive. We previously reported that *T. gondii*, unable to synthesize cholesterol, is auxotrophic for plasma LDL-cholesterol internalized into the host cell [1,2]. The parasite diverts LDL-loaded endocytic organelles to the PV to encroach them into PV membrane (PVM) invaginations, then trap them to the vacuolar space to retrieve their cholesterol content [3,4]. *Toxoplasma* expresses an ATP-binding cassette (ABC) G family transporter localized at its plasma membrane that acts as a cholesterol importer [5]. Within the parasite, a bifunctional protein containing two sterol-carrier protein-2 domains promotes the circulation of cholesterol (in addition to phospholipids and fatty acids) between organelles and the plasma membrane [6]. Host-derived cholesterol is inserted in the parasite plasma membrane and organelles, with special enrichment in the secretory organelle rhoptries [1,7,8], involved in PVM formation [9]. When LDL are provided in excess in the culture medium, the parasite salvages large amounts of cholesterol from LDL and esterifies this lipid using acyl-CoA:cholesterol acyltransferase (ACAT) enzymes for storage as cholesteryl esters in lipid droplets [6,10,11].

Apart from these studies informing on the source, trafficking and storage of cholesterol in *Toxoplasma*, nothing is known about the utilization of cholesterol by the parasite. In eukaryotic cells, cholesterol contributes to the structural integrity and fluidity of the bilayer, and thus to the function of transmembrane proteins. In addition, cholesterol fulfills critical metabolic functions, serving as a precursor for the synthesis of steroid hormones, oxysterols, bile acids and vitamin D. Like any eukaryotic organism, *Toxoplasma* incorporates cholesterol into its membranes for bilayer organization but whether *Toxoplasma* has enzymes for the break down and consumption of cholesterol, or utilizes this lipid as building blocks for the production of cholesterol-derived metabolites such as steroids, remains to be elucidated.

In vertebrates, steroid hormones belong to five major classes: testosterone, estradiol, progesterone, cortisol/corticosterone (glucocorticoid), and aldosterone (mineralocorticoids), and they play important reproductive and developmental roles. Invertebrates synthesize many steroid molecules such as ecdysteroids that have hormonal roles involved in the regulation of ecdysis and development [12]. Steroids are also integral components of plants where they are synthesized *de novo* from phytosterols. Plant steroids including progesterone, testosterone, androstadienedione, androstenedione and estrogens act as chemical messengers for cell-cell communication and are required for the regulation of plant growth, development, and reproduction [13].

In mammalian cells, the synthesis of cholesterol-derived metabolites involves the activities of the cytochrome P450 family of oxidases (CYP450s) [14]. The activation of cholesterol-metabolizing CYP450s often requires the donation of electrons from hemoproteins, such as the NADPH-dependent cytochrome P450 reductase (CPR) in the ER, which shuttles electrons from NADPH through the FAD and FMN-coenzymes into the iron of the prosthetic heme-group of the CYP450. In some cases, activation of cholesterol-metabolizing CYP450s occurs through interaction with heme-binding proteins containing a cytochrome b5 (cytb5) domain. Among them are the members of the membrane-associated progesterone receptor (MAPR) family including progesterone receptor membrane component-like proteins (PGRMC1 and PGRMC2), neudesin (NENF) and neuferricin (NEUFC) [15–17]. In mammalian cells, PGRMC1 proteins localize to the plasma membrane, endoplasmic reticulum (ER), endosomes, nucleus and extracellular environment, and fulfill a broad range of functions related to cell survival, including cholesterol synthesis and homeostasis, steroid hormone production and signaling, and resistance to DNA damage-induced stress [18]. PGRMC1 is overexpressed in malignant cancer cells and localizes to mitochondria [19,20]. In *Saccharomyces cerevisiae* and *Schizosaccharomyces pombe*, the PGRMC1 homologue named damage-associated protein 1 (Dap1), localizes to the ER and endosomes and is involved in ergosterol synthesis and resistance to DNA damage-induced toxicity due to hypoxia and drug exposure [21,22].

Steroidogenesis entails enzymatic steps by which cholesterol is converted to biologically active steroid hormones. Steroidogenic enzymes fall into two groups: cytochrome P450 enzymes (type 1 in mitochondria and type 2 in the ER) and hydroxysteroid dehydrogenases [23]. The initial step in steroidogenesis is the conversion of cholesterol to pregnenolone by the cytochrome P450 side-chain cleavage enzyme in mitochondria [24]. A search of the genomic database of *T. gondii* (www.toxoDB.org) reveals that the parasite genome contains one single gene coding for a type 1 CYP450 enzyme (TGME49_315770) and one single gene encoding a mitochondrial MAPR homolog (TGME49_276990). *Toxoplasma* has also several genes encoding for zinc finger HIT domain-containing proteins known to interact with nuclear hormone receptors (e.g., TGME49_212810 in the nucleolus) [25], and genes coding for enzymes with potential activities in steroid transformations, such as 3-oxo-5-alpha-steroid 4-dehydrogenase that converts testosterone into 5-alpha-dihydrotestosterone and progesterone or corticosterone into their corresponding 5-alpha-3-oxosteroids (TGME49_272180; TGME49_285240 and TGME49_304480). Finally, the parasite genome contains 3 genes for non-mitochondrial (microsomal or nuclear) cytochrome b5 family heme/steroid binding domain-containing proteins (TGME49_240770; TGME49_276110 and TGME49_313580).

In this study, we have investigated the physiological roles of mitochondrial CYP450 enzyme and MAPR homolog of *T. gondii* to examine whether the parasite has steroidogenic potential. Our targeted lipidomics analysis detects the presence of selected hormone steroids in *T. gondii*. Functional characterization of the CYP450 and MAPR homologs reveals their dynamic interaction in the parasite mitochondrion, and decreased expression of each of these proteins correlates with reduced steroid amounts in *Toxoplasma*.

Current research on pathogenic infections underscores the exploitability of cholesterol scavenging and utilization for therapeutic interventions [26–29]. *Toxoplasma gondii* is a leading opportunistic parasite in immunosuppressive conditions. Providing a comprehensive view on cholesterol homeostatic and biosynthetic pathways in *T. gondii*, extended to unique steroidogenic mechanisms, may expose new vulnerabilities.

## Results

### *Toxoplasma* contains selective classes of steroid hormones

In mammalian cells, steroidogenesis is a multi-enzymatic process initiated by the translocation of cholesterol from intracellular stores into mitochondria to generate pregnenolone. Pregnenolone enters the endoplasmic reticulum (ER) where further enzymatic reactions occur to form steroids that return to the mitochondria to be transformed into major steroid hormones. To examine whether *Toxoplasma* contains steroids, we conducted LC-MS steroid content analysis of purified extracellular parasites, using steroid standards. Among 24 steroids analyzed, 14 were detected in *Toxoplasma* and quantitative measurement revealed their presence at various amounts, with the highest concentrations for hydroxylated pregnenolone, dehydroepiandrosterone (DHEA) and deoxycorticosterone (DOC) (Table 1 and S1 Fig). No steroids were detected in host fibroblast debris and culture medium.

### *Toxoplasma* contains a single mitochondrial TgCYP450 homolog with a conserved heme iron-binding domain

PGRMC1/Dap1 proteins activate many cholesterol-metabolizing CYP450s and steroidogenic enzymes, including CYP11A1 (involved in the conversion of cholesterol to pregnenolone, the precursor of all steroid hormones), CYP21A2 (involved in biosynthesis of aldosterone and cortisol), CYP17 (a bifunctional enzyme with 17α-hydroxylase and 17,20-lyase activities, implicated in the conversion of pregnenolone to 17OH-pregnenolone and progesterone for androgen and glucocorticoid syntheses), CYP19 (involved in estrogen biosynthesis), CYP51A1 and CYP61A1 (two lanosterol-14-demethylases) [30,31]. The genome of *T. gondii* contains a gene (TGME49_315770; www.toxoDB.org) encoding a protein (predicted molecular weight: 62.2 kDa) that carries hallmark features of cytochrome P450 enzymes. with sequence similarity ranging from 26–40% with human cytochrome P450 (S2A Fig). The protein has a conserved heme iron-binding domain [FGFGTRKCLG] (consensus pattern: [FW]-[SGNH]-x-[GD]-{F}-[RKHPT]-{P}-C-[LIVMFAP]-[GAD]), a transmembrane domain and a predicted mitochondrial localization based on hyperLOPIT (www. ToxoDB.org). However, the highest similarity of the *T. gondii* CYP450 (TgCYP450) is with the sequence of microsomal CYP26B1 involved in the biosynthesis of retinoic acid. TgCYP450 shares sequence similarity with six mitochondrial CYP450 involved in steroidogenesis in mammalian cells: CYP27B1 (involved in the conversion of vitamin D to its active form, 1,25-dihydroxyvitamin D3); CYP11B1 (involved in the conversion of 11-deoxycortisol to cortisol); CYP21A2 (involved in the conversion of progesterone and 17-hydroxyprogesterone to 11-deoxycorticosterone and 11-deoxycortisol); CYP17A1 (an 17α-hydroxylase involved in the synthesis of progestogens: 17OH-pregnenolone, 17OH-progesterone, dehydroepiandrosterone, androstenedione, 16OH-progesterone); CYP11A1 (also named CYP450 side chain cleavage (scc) enzyme, involved in the conversion of cholesterol to pregnenolone); and CYP11B2 (also named aldosterone synthase, involved in 3 sequential reactions to produce aldosterone via the conversion of 11-deoxycorticosterone to corticosterone, to 18-hydroxycorticosterone, then to aldosterone). TgCYP450 shares high sequence identity with CYP450 homologs present in fellow members

**Table 1. Detection of selective steroids in *Toxoplasma*.**

| Steroids present | ng/g of parasites |
|---|---|
| Pregnenolone | 0.1198 |
| 17-hydroxypregnenolone | 10.02 |
| 17α-hydroxypregnenolone | 2.095 |
| Progesterone | 0.4902 |
| 11-deoxycorticosterone (DOC) | 2.057 |
| Corticosterone | 0.1914 |
| 11-deoxycortisol | 0.0326 |
| Cortisone | 0.1501 |
| Aldosterone | 0.0161 |
| Dehydroepiandrosterone (DHEA) | 3.0311 |
| 5α-androstanediol | 1.457 |
| 4-androstene-3,17-dione | 0.8192 |
| *trans*-androsterone | 0.7075 |
| Dihydrotestosterone | 0.3212 |
| Hydrocortisone | ND |
| Testosterone | ND |
| β-nortestosterone | ND |
| *cis*-androsterone | ND |
| 19-norandrostenedione | ND |
| Estriol | ND |
| Estradiol | ND |
| 19-norandrostenedione | ND |
| 2-methoxyestradiol | ND |
| Estrone | ND |

LC-MS steroid content analysis of purified extracellular *Toxoplasma* compared to steroid standards. Additional steroid detection assays included supernatant from host cell lysis containing host cell derived metabolites and debris for which no steroids were detected. Data are means of two parasite samples. See details of regression equations for the tested steroids in S1 Fig. ND: non detected.

of the Sarcocystidae family: 97% with *Hammondia hammondi*, 80% with *Neospora caninum*, 72% for *Besnoitia besnoiti*, 63% for *Cystoisospora suis* and 57% for *Sarcocystis neurona* (S2B Fig). These CYP450 homologs have a C-terminal region harboring a helix K domain to stabilize the protein core, and a heme-binding loop to position the iron atom in the heme.

## TgCYP450 localizes to the parasite mitochondrion

We analyzed the localization of TgCYP450 in intracellular *Toxoplasma*. Using the CRISPR/Cas9 system, we engineered a *Toxoplasma* cell line for endogenous tagging of the *CYP450* gene in fusion with HA (S3A Fig). IFA on intracellular TgCYP450-HA-expressing *Toxoplasma* using anti-HA antibody illustrated a tubular fluorescent pattern that overlapped with the Mito-Tracker signal (Fig 1A). A double IFA using an antibody against HA and the mitochondrial marker HSP70 [32] was performed on PV of different sizes containing TgCYP450-HA-expressing *Toxoplasma*, and data showed colocalization, confirming the distribution of TgCYP450 in the mitochondrion of *Toxoplasma* (Fig 1B). Measurements of the Pearson's correlation coefficients (PCC) and Mander's overlap coefficients (MOC) show a strong positive correlation between the HA and HSP70 signals regardless of the PV size. We therefore named this protein, TgCYP450mt.

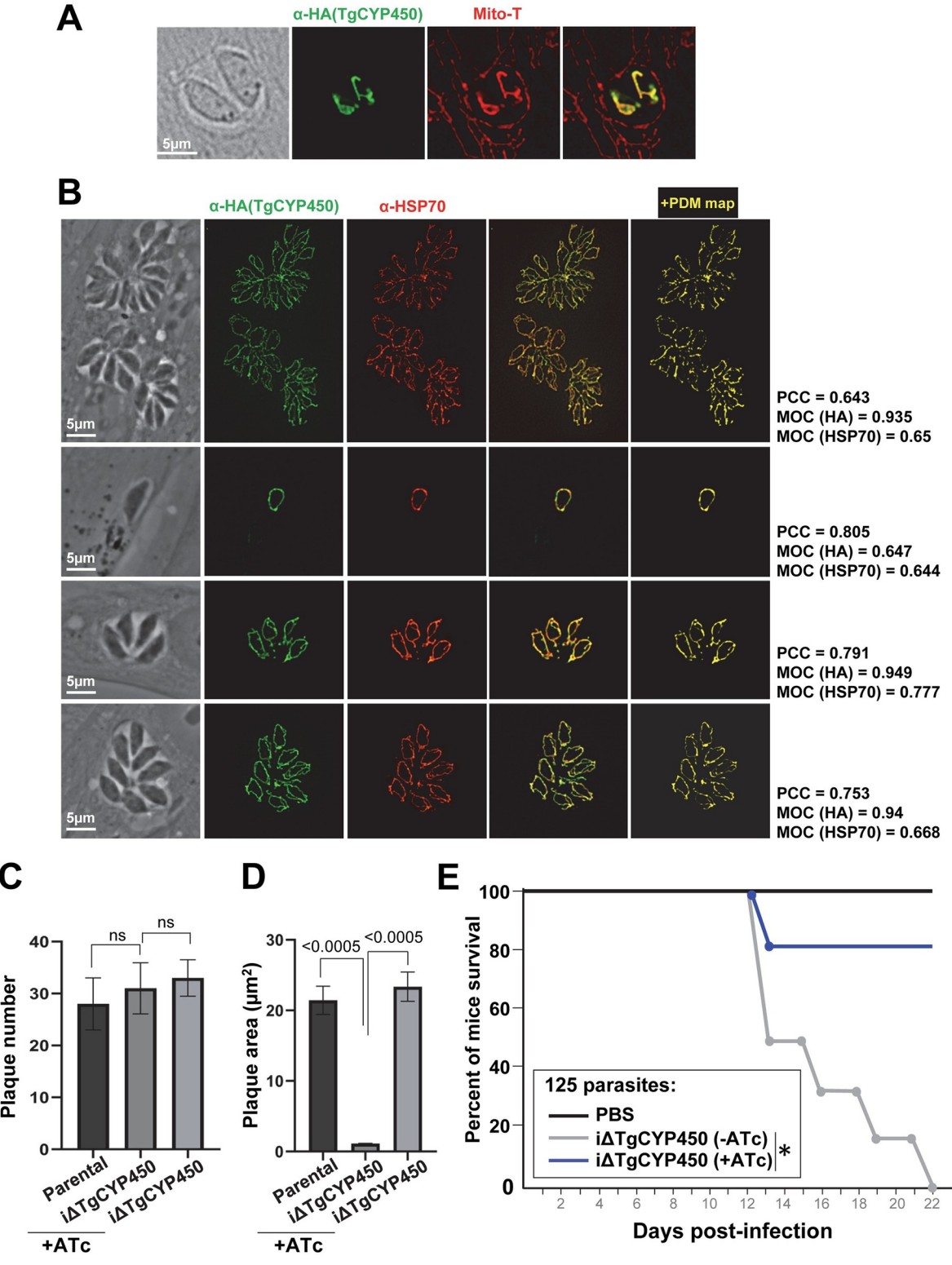

**Fig 1. Localization of TgCYP450 in *Toxoplasma* and phenotypic analysis of CYP450 conditional KO (iΔTgCYP450) parasites.** (A) IFA on TgCYP450-HA-expressing *Toxoplasma* using anti-HA antibody showing costaining with Mito-Tracker. (B) Double IFA on TgCYP450-HA-expressing *Toxoplasma* using anti-HA and anti-HSP70 antibodies showing colocalization. Values of PCC and MOC are shown. (C-D) Plaque assays using parental and iΔTgCYP450 parasites on HFF monolayers to monitor invasion in C: intracellular parental and iΔTgCYP450 tachyzoites were exposed for 3 days to 1 μg/ml ATc (or solvent for iΔTgCYP450 control). Egressed parasites were

collected from these cultures and 1,500 parasites were used for plaque assays before counting of the plaques after 4 days. Plaque assays using parental or iΔTgCYP450 parasites on HFF monolayers to assess growth in D: monolayers infected with 125 parental or iΔTgCYP450 parasites for 8 days in medium containing 1 μg/ml ATc (or solvent). (E) Virulence assays: i.p. injection of 125 iΔTgCYP450 or iΔTgCYP450 parasites pre-exposed to ATc or solvent *in vitro*, or PBS in Swiss-Webster mice (6 per group). One group of mice (blue line) was given drinking water with ATc. The mortality of mice was monitored daily until Day 22. *, $p = 0.0001$ (Log-rank Mantel-Cox test).

## An induced conditional KO for TgCYP450mt has severe growth defects in vitro and is poorly virulent

To investigate the physiological relevance of TgCYP450mt for *Toxoplasma* development and pathogenicity, we attempted to engineer a parasite cell line lacking the *CYP450* gene by double recombination; however, the parasites lacking TgCYP450 could not be propagated in culture, suggesting that the *CYP450* gene is essential. We therefore created a conditional knockout (cKO) of the *cyp450* gene through the displacement of the endogenous promotor with an inducible promotor under the control of anhydrous tetracycline. Promotor replacement was performed in the C-terminally 3xHA-tagged parasite line (S3A Fig) for stable expression of an ATc-responsive transactivator protein in a TATi ΔKu80 background strain (S3B Fig). Conditional expression of TgCYP450mt upon ATc addition for 24 h or 48 h was verified on Western blots using anti-HA antibody.

Plaque assays were performed to assess the ability of the CYP450cKO (or iΔTgCYP450) parasites under ATc to invade (based on plaque numbers) and to grow (based on plaque sizes). Compared to parental parasites exposed to ATc and iΔTgCYP450 without ATc added, iΔTg-CYP450 parasites treated with ATc showed no invasion defects (Fig 1C) but their growth was impaired by ~90% (Fig 1D). Virulence assays were performed by intraperitoneal infection of outbred mice. Data showed that 80% of mice infected with iΔTgCYP450 parasites exposed to ATc remained alive at Day 22, as opposed to all mice infected with iΔTgCYP450 parasites without ATc that died (Fig 1E). These observations indicate that TgCYP450mt contributes to parasite development *in vitro* and survival in mice.

## *Toxoplasma* contains one TgMAPR homolog with conserved residues in a heme-binding cytochrome b5 domain

Members of the MAPR family are functional partners of many members of the CYP450 system to influence the enzymatic CYP function [33]. These proteins are widespread in eukaryotes, evolutionarily conserved and distant homologs of the hemoprotein cytochrome b5 reductase (CYP5) [16,34]. MAPR have in common a non-covalent heme-binding domain made of five-coordinated heme iron involving a tyrosine, as opposed to the six-coordinated heme with two axial histidines in CYP5 [35–37], allowing for the formation of homodimers through hydrophobic heme-heme stacking interactions. These proteins have no homology with steroid receptors and are not directly involved in progesterone binding, instead exerting their function at a non-genomic level (i.e., not involving changes in gene expression), in pathways related to steroid hormone production and resistance to DNA damage [16,18]. Phylogenetic analyses of MAPR-related proteins in eukaryotes reveal that genes encoding these proteins are absent in many parasitic protists [18], perhaps related to selective parameters associated with a parasitic mode of life. Among the phylum of apicomplexan parasites, only some members of the Sarcocystidae family (*Toxoplasma*, *Neospora*, *Hammondia*, *Besnoitia*) possess an ancestral MAPR gene with a C-terminal cytochrome *b5* family heme/steroid binding domain (S4A Fig). Compared to the human PGRMC1, the archetypal MAPR member, and to yeast Dap1, the MAPR sequence in *T. gondii* (TGME49_276990; www.toxoDB.org) shows 31% and 34% identity, respectively. The parasite sequence has a predicted mitochondrial localization based on

hyperLOPIT [38] and harbors one predicted transmembrane domain at the N-terminus (https://cctop.ttk.hu/) with conserved residues (Y151, Y157 and D164 corresponding to Y107, Y113 and D120 in PGRMC1) required for heme binding (S4B Fig) [18,39,40]. The parasite sequence contains additional conserved residues with PGRMC1. For example, the T74 and T101 phosphorylation sites of human PGRMC1, which exhibit deep phylogenetic conservation, are present in the *T. gondii* sequence at T75 and T142. The *T. gondii* sequence contains the phosphate acceptors (Y229 and S230) of the predicted Src homology 2 (SH2) domain target motifs (Y180 and S181 in PGRMC1), thought to mediate inducible protein interactions with SH2 domain-containing proteins. A phylogenetic study reported that the signaling phosphorylated tyrosine Y180 adjacent to a D/E region (DE182-183) and a tandem positive charge (RK192-193) motif present in the human PGRMC1 sequence and conserved among PGRMC-like proteins, may have appeared with the emergence of the eumetazoan common ancestor (the Urmetazoan) [17]. Intriguingly, the *Toxoplasma* sequence contains a tyrosine (Y229) close to an EE region (EE231-232) that is adjacent to an arginine and lysine (RK192-193). However, the Y229 phosphorylation required for the activation of signaling pathways, needs to be verified before inferring the evolutionary origin of PGRMC1 homologs within the protist lineage. A putative SH3 domain binding motif is also present in the parasite sequence but not the CK2 motif. Finally, while PGRMC proteins possess the prominent residues F106, P109 and P112 (as in PGRMC1) around the heme-binding pocket for heme binding, only 2 of the 3 are detected in the *Toxoplasma* sequence (F150 and P156). Based on these phylogenetic analyses, we therefore named this *Toxoplasma* protein TgMAPR. The TgMAPR sequence shares 93%, 72% and 62% identity with homologous sequences in *H. hammondi*, *N. caninum* and *B. besnoiti*, respectively; only, the *N. caninum* MAPR contains the 3 conserved residues around the heme-binding pocket (F153, P156 and P159) present in all other PGRMC proteins.

## TgMAPR is a membrane-associated hemoprotein

The primary sequence of TgMAPR has a predicted molecular weight of 26.2-kDa, a transmembrane domain and the cytochrome b5 family heme-binding domain found in all MAPR family members. It has been reported that in the presence of heme, PGRMC1 forms a dimeric structure largely through hydrophobic interactions between the heme moieties of two monomers [37]. We analyzed the expression of TgMAPR in *Toxoplasma* and its physical properties. We generated a recombinant TgMAPR peptide in *E. coli* (6xHis-TgMAPR) for anti-TgMAPR antibody production in mice. Western blotting on freeze-thaw lysates of WT parasites using anti-TgMAPR antibody showed a major band of 27-kDa, corresponding to the mass of TgMAPR, in addition to a weaker upper band at 52.5-kDa, likely corresponding to a TgMAPR dimer (Fig 2A). To assess if TgMAPR is a transmembrane protein, we extracted the parasite proteins with either the detergent Triton X-100 or sodium carbonate at pH 11.5. The protein was detected in the soluble fraction in Triton X-100 extraction but in the membrane fraction upon $Na_2CO_3$ extraction, similarly to the integral membrane protein TgABCG [5] and distinctly from the soluble GRA1 protein (Fig 2B). We conclude that TgMAPR is a membrane protein as predicted in the sequence.

We next analyzed the heme-binding properties of TgMAPR using three approaches. First, heme-reconstituted recombinant TgMAPR was subjected to a whole spectrum absorbance scan from 350 to 600 nm, before and after addition of the oxidizing agent potassium ferricyanide. Data in Fig 2C showed a major peak at 440 nm that shifted to 425 nm after oxidation, a hallmark of heme-containing proteins [35]. Second, chemiluminescence is a very sensitive method for detecting heme-containing proteins in electrophoretic gels [41] and TgMAPR heme-binding was further examined in an in-gel heme assay by visualization of the intrinsic

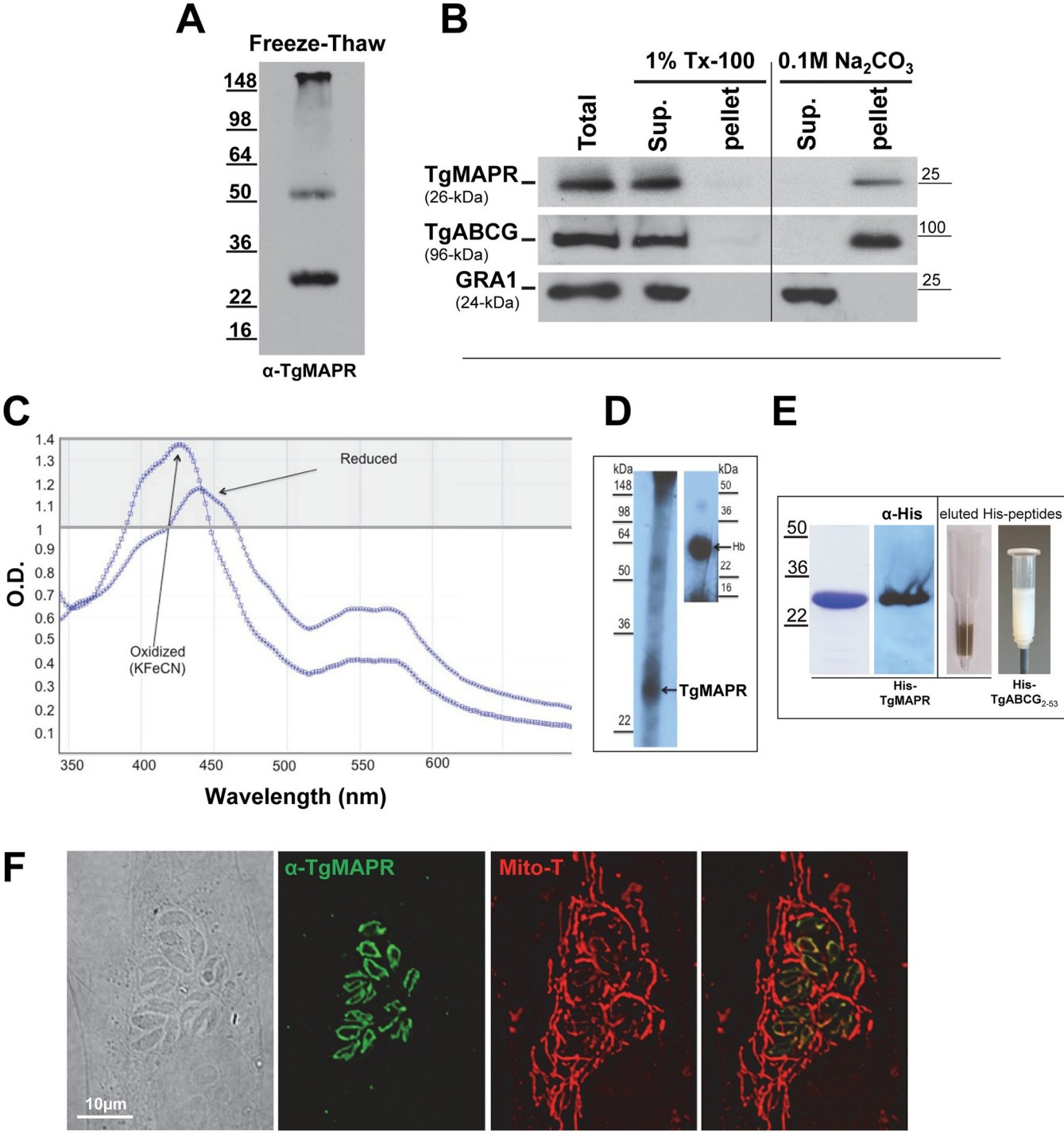

**Fig 2. Membrane-association, heme binding properties and localization of TgMAPR.** (A) Western blots on freeze-thaw lysates from WT parasites using anti-TgMAPR antibody revealing 2 bands corresponding to monomeric and dimeric forms of TgMAPR. (B) Western blotting analysis of WT parasites with proteins extracted by 1% Triton X-100 (TX-100) or sodium carbonate ($Na_2CO_3$) showing TgMAPR partitioning into the pellet phase of the $Na_2CO_3$ extraction fraction, like the integral ER transporter TgABCG and unlike soluble GRA1 detected in the supernatant (Sup.) phase of the carbonate extract. (C) Absorption spectra of recombinant TgMAPR, showing the spectra of reduced and oxidized form after the addition of KFeCN. (D) In-gel heme assay on recombinant TgMAPR and hemoglobin (Hb) as positive control detected by ECL solution. (E) Coomassie blue and anti-6xHis western blot of purified 6xHis-TgMAPR plus a photo of the recombinant protein eluted from the purification column showing the recombinant 6xHis-TgMAPR protein colored brown due to presence of the heme moiety, in contrast to purified peptide 6xHis-TgABCG$_{2-53}$ eluted from a $Ni^{2+}$ resin column as a white suspension. (F) IFA using anti-TgMAPR antibody and Mito-Tracker (Mito-T) labeling of WT showing overlap in the parasite mitochondrion.

peroxidase activity of the heme group [42]. Recombinant TgMAPR was run on an SDS-PAGE gel and stained by enhanced chemiluminescence prior to exposure to X-ray film; the 26-kDa band of recombinant TgMAPR gave a positive signal (Fig 2D). Third, we observed that purified recombinant TgMAPR bound to the purification column, was tinged with brown (Fig 2E), which is a characteristic of 5-coordinate hemoproteins like PGRMC1, in contrast to the red color of 6-coordinate cytochrome *b5* heme proteins like CYP5 [42] and white color for not hemoprotein (shown here with TgABCG).

These data establish that TgMAPR is associated with parasite membranes either as monomers or dimers, and binds heme.

## TgMAPR localizes to the mitochondrion

MAPR family members have various subcellular localizations, according to their functions. We next analyzed the distribution of TgMAPR in intracellular *Toxoplasma* by IFA using anti-TgMAPR antibody. We observed a fluorescent lasso-shape signal around the parasite nucleus (Fig 2F) that is reminiscent of the shape of the mitochondrial network in the parasite. Parasites were then co-stained with Mito-tracker and anti-TgMAPR antibody, and the two fluorescent signals largely colocalize. In parallel, we examined the distribution of TgMAPR expressed in HeLa cells upon transfection with a plasmid containing TgMAPR-HA, and IFA using anti-HA antibody showed the fluorescent signal in mitochondria identified with Mito-Tracker (S5A Fig). To confirm the localization of TgMAPR in the mitochondrion, we engineered a parasite stable line expressing TgMAPR-HA; western blotting analysis showed expression of TgMAPR-HA as a monomer at 27.5-kDa and dimer at 53-kDa (S5B Fig) as observed for WT parasites (Fig 2A). Double IFA of TgMAPR-HA-expressing parasites using anti-HA and anti-MAPR antibodies showed colocalization for signal specificity (S5C Fig) and immunoEM gold staining using anti-HA antibody detected gold particles on mitochondrial membranes (S5D Fig). Finally, we performed double IFA on this parasite strain using anti-HA and anti-HSP70 antibodies, and data show colocalization, with positive values ($> 0.5$) for PCC and MOC (S5E Fig). In few parasites (~5%) some area of the mitochondrion appeared more enriched in TgMAPR than in HSP70 (S5F Fig).

## TgMAPR functionally complements *S. pombe* dap1Δ mutant

We next wanted to functionally validate TgMAPR as a canonical MAPR family member. In fission yeast, SpDap1 plays a role in ergosterol biosynthesis and yeast survival under hypoxia conditions through interaction with the sterol-synthesizing cytochrome P450 protein Erg11p/Cyp51p; Schizosaccharomyces pombe Dap1 or SpDap1expression is stimulated by low oxygen/sterol concentrations [21,43]. As TgMAPR and Dap1 share conversed residues including the Cytb5 domain (S4B Fig), we analyzed the functional equivalence of TgMAPR and Dap1. We expressed TgMAPR in *S. pombe* lacking *Dap1* to examine the potential ability of TgMAPR to restore Dap1 activities in mutant yeast, related to resistance to hypoxia and sterol production. Transformation of *SpDap1Δ* with TgMAPR with a *myc* tag at the C-terminus resulted in heterologous expression of TgMAPR in yeast (S6A Fig, panel i). Parental, *SpDap1Δ* and complemented strain with HA-SpDap1 or TgMAPR were exposed to CoCl$_2$, which mimics hypoxic conditions by activating the hypoxia-inducible factor 1α (HIF1α) [44]. We observed growth rescue in two independent clones of mutant yeast expressing TgMAPR to the same extent as for parental yeast and *SpDap1Δ* complemented with HA-SpDap1 (S6A Fig, panel ii). This suggests that TgMAPR and SpDap1 similarly protect yeast from hypoxic damage.

In yeast, Dap1 contributes to ergosterol synthesis, indeed compared to WT *S. pombe*, *SpDap1Δ* has reduced amounts of ergosterol and elevated amounts of the ergosterol

biosynthetic intermediates: 24-methylene lanosterol, ergosta-5,7,24(28)-trienol, and ergosta-5,7-dienol, consistent with defects at the Erg11 and Erg5 enzymatic steps (see pathway in S6B Fig) [21,43]. Heterologous expression of TgMAPR in *Spdap1Δ* resulted in the rescue of ergosterol synthesis to a similar extent as *SpDap1Δ* complemented with HA-SpDap1 (S6B Fig). These data indicate that TgMAPR can analogously replace Dap1 in yeast for hypoxia protection and ergosterol synthesis, however, they do not imply similar functions of TgMAPR in *Toxoplasma* more especially since this parasite has no sterol biosynthetic machinery.

## TgMAPR-deficient *Toxoplasma* propagate poorly in vitro and suffer from a fitness loss

PGRMC1/Dap1 is associated with increased cell survival, more especially under stress conditions [21,43,45]. To investigate the physiological importance of TgMAPR, we disrupted the *mapr* gene via replacement with a DHFR resistance marker in the *T. gondii* ΔKu80/RH strain (S3C Fig). Few viable KO clones could be obtained and they exhibited very slow growth. We quantified the growth rate of ΔTgMAPR in cultured cells by plaque assays for 7 days allowing several cycles of parasite invasion requiring motility, replication and egress. Measurement of plaque area showed significant differences between the KO and parental parasites with ~7-times smaller plaques formed by the mutant (Fig 3A). Next, we examined which steps in the parasite lifecycle could be affected upon TgMAPR deficiency. For host cell invasion, *Toxoplasma* employs a form of gliding motility that depends on its actomyosin system [46]. The motility of the parasites can be visualized by trails of surface proteins (e.g., SAG1) deposited on a coated surface and revealed by IFA. We performed trail deposition assays allowing ΔTgMAPR and parental parasites to glide for 30 min with IFA using anti-SAG1 antibody. Circular SAG1-containing trails were observed for both strains with no difference in their length (Fig 3B). To assess the ability of ΔTgMAPR to invade mammalian cells, we used the red/green invasion assay to discriminate between parasite solely attached to the host cells versus fully internalized [47]. Data illustrate significant defects in invasion for the mutant with an almost 2-fold reduction of host cell penetration events compared to parental parasites, with 5-times more mutants still attached to the host cell surface (Fig 3C). Quantification of replication rates of ΔTgMAPR using [$^3$H]uracil incorporation assays revealed ~20% less radioactivity associated with the mutant compared to parental parasites (Fig 3D). Next, we examined the ability of ΔTgMAPR parasites to exit from their host cells. A rise in calcium concentration is associated with the secretion of proteins involved in rapid egress of *Toxoplasma* from the host cell [48]. Parasite egress was chemically induced by the calcium ionophore A23187 and visualized by time-lapse microscopy to record time until egress for ΔTgMAPR and parental parasites. At 100 sec post-induction, many PV containing ΔTgMAPR parasites were still intact, in contrast to PV for the parental strain that were all lysed (Fig 3E); data showed a ~3-fold egress delay for the mutant. Jointly, these observations indicate poor fitness of ΔTgMAPR parasites at various steps in their developmental cycle *in vitro*.

## ΔTgMAPR parasites exhibit severe endodyogeny defects, with an abnormally long S-phase in the cell cycle

*Toxoplasma* parasites divide by a process of endodyogeny during which two daughter cells are synchronously assembled within the mother cell every 7–8 hours, resulting in geometric expansion of clonal progeny until host cell lysis ∼ 48 h [49]. The significant growth defects observed for ΔTgMAPR parasites may be associated with severe cytopathies in the ultrastructural organization of these mutants. ΔTgMAPR parasites were cultivated for 24 h in fibroblasts for EM inspection. A striking feature was the accumulation of dense granular material within

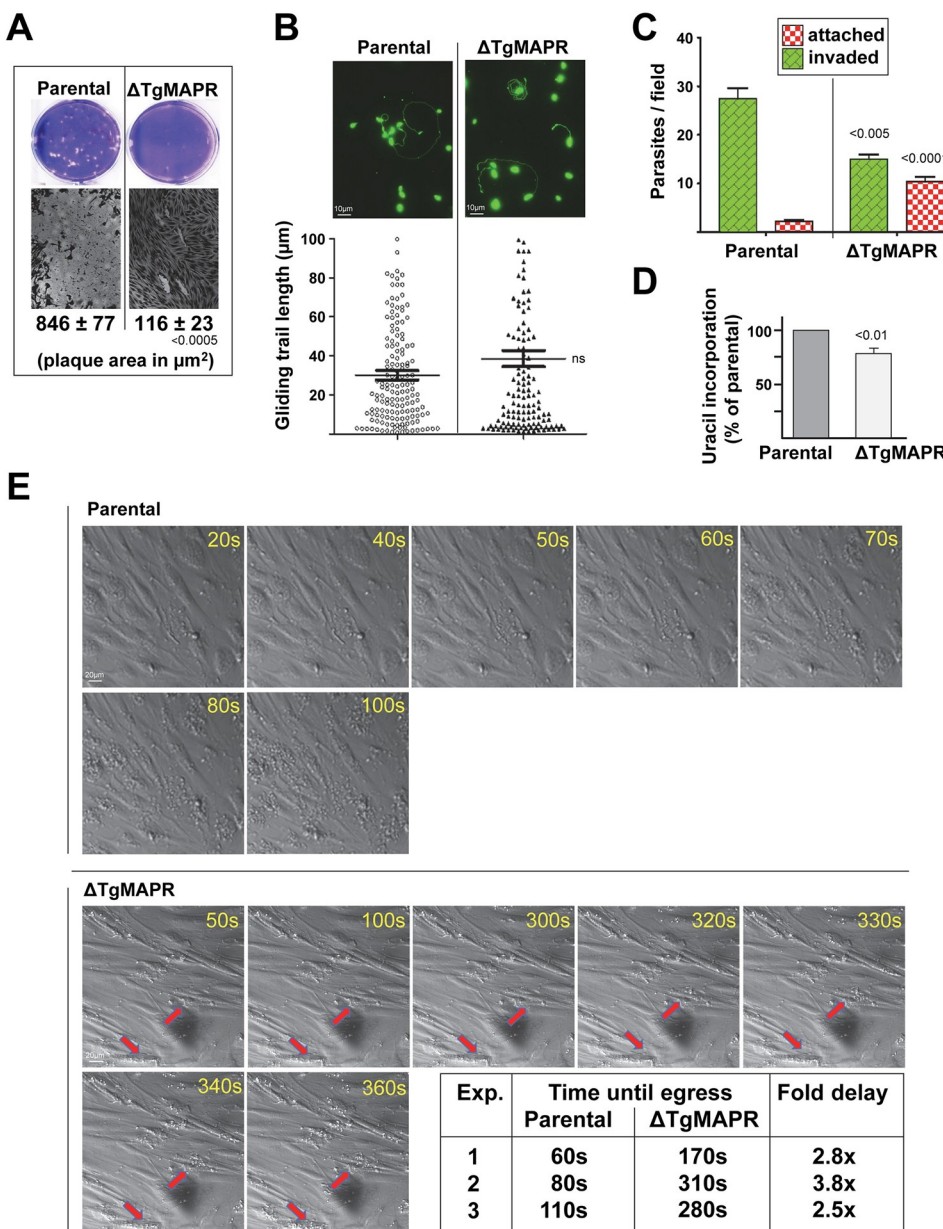

**Fig 3. Phenotypic analysis of ΔTgMAPR parasites.** (A) Plaque assays. Analysis of parasite growth for 7 days after fibroblast infection with 150 parasites from the parental and ΔTgMAPR strains showing representative images and quantification of lysed area from 3 independent experiments. Data are means ± SD. (B) Gliding assays. Freshly lysed ΔTgMAPR1 and parental parasites were allowed to glide on FBS-coated glass slides for 30 min before fixation and staining with anti-SAG1 antibody under non-permeabilized conditions. Representative images are shown. Measurement of gliding trails by dot plots, showing means ± SD of 3 independent experiments. (C) Invasion assays. Quantification of invasion of parental and ΔTgMAPR parasites using the red/green invasion assay. Red histograms represent external, attached parasites while green histograms represent internal, penetrated parasites. Data are means ± SEM of 4 independent experiments. (D) [³H]uracil incorporation assays for replication. HFF were infected with parental and ΔTgMAPR parasites for 24 h prior to incubation with tritiated uracil. Data are means ± SD of 3 independent experiments. (E) Induced egress assays. HFF were infected with parental and ΔTgMAPR parasites for 24 h prior to exposure to 2 μM A23187 for time-lapse microscopy views, showing rapid egress for parental parasites that were largely extracellular upon treatment and slower response for the treated mutant (arrows). Quantification of egress time with means in seconds for 3 treated monolayers infected with parental or ΔTgMAPR parasites.

all PV in which ΔTgMAPR parasites seemed embedded (Fig 4A). On some sections, the parasites exhibited a 'gaping hole' at the basal end as a sign of defective membrane sealing during the formation of the daughter cells (Fig 4A, arrow in panel i), suggesting discharge of parasite material into the PV. The PV of WT tachyzoites contains an Intravacuolar Network (IVN) of membranous tubules that is initially secreted by the parasite [50], then expanded through the uptake of host lipid uptake [51]. One function of the IVN is trapping host cytosolic proteins and host organelles as sources of nutrient [4,52]; strikingly, no IVN was observed in the ΔTgMAPR PV lumen. WT intracellular tachyzoites usually contain 1 to 3 lipid droplets [6,10]. In contrast, several large lipid droplets were observed in ΔTgMAPR parasites, up to 4 per section (Fig 4A, panels i to iii), suggesting important storage activities of cholesteryl esters and triglycerides or impairment in using these neutral lipids for metabolic functions. Amylopectin granules are energy reserves that fuel the transition from proliferative tachyzoites to slow-growing bradyzoite cysts [53]; however, in ΔTgMAPR tachyzoites, abundant amylopectin granules were visible, up to 15 per parasite section (Fig 4A, panels ii and iv), as a sign of metabolic stress. The morphology of several organelles was also unusual, such as the mitochondrion, that was enlarged with disorganized cristae (Fig 4A, panel iv and inset) and rhoptries showing a high electron-density, suggesting the accumulation of osmiophilic material (e.g., lipids). While some replication profiles with nascent daughters were observed with 2 or 4 individual parasites per PV (Fig 4A, panel iii and iv), more often a mass of poorly differentiated parasites intertwined with each other was noticed (Fig. 4A, panel v). Like WT parasites, ΔTgMAPR parasites were able to recruit host mitochondria at the PV (Fig 4A), indicating functional secretory activities to export the mitochondrial association factor 1 (MAF1) from dense granules to the PV membrane [54].

Cyst forms of *Toxoplasma* are characterized by a thick glycosylated cyst wall derived from the PV membrane detectable in a *Dolichos biflorus* Agglutinin (DBA) lectin fluorescence binding assay [55]. Due to the slow replication rate of ΔTgMAPR parasites, we performed the DBA assay to determine whether these mutants were converting to cyst forms. At 40 h p.i., we detected a strong DBA signal all around most (97%) ΔTgMAPR PV, in contrast to parental PV with either no or a weaker lectin staining in only 6% of PV (Fig 4B). However, IFA using antibody against the protein bradyzoite marker BAG1/HSP30 did not reveal any fluorescence signal. In addition, our RNA-Seq analysis of ΔTgMAPR parasites did not show any downregulated tachyzoite or upregulated bradyzoite transcripts (see below S8 Fig). This indicates that deletion of the *mapr* gene does not induce the differentiation of tachyzoites to bradyzoites, but just results in major stress and poor fitness of the mutant.

*Toxoplasma* tachyzoite replication differs from the classic animal cell cycle as it is characterized by an interwoven relationship between mitosis and cytokinesis driven by the internal budding process of daughter cells. The tachyzoite cell cycle has a bimodal distribution, with a major 1 N DNA peak (= G1 phase for ~60% of the parasite population) and a major 2 N DNA peak (encompassing S, G2 phase and mitosis for ~30% of the parasite population; in between, there is an early-mid S phase (1–1.6 N for ~10% of the parasite population) [56]. We next conducted flow cytometry analyses on ΔTgMAPR parasites isolated from VERO cells to examine their cell cycle profiles, in comparison with parental parasites. Data show that ΔTgMAPR parasites had an increased S-phase compared to parental parasites (Figs 4C and S7A), compatible with slow daughter cell budding (Fig 4A). Based on three independent preparations of parasites, the percentages of control parasites were 63.7 ± 3.4% in G1 interphase and 32.0 ± 1.9% in S/G2/M phases while the percentages of ΔTgMAPR parasites were 31.5 ± 4.2% in G1 interphase and 67.9 ± 3.6% (*, p<0.005) in abnormal phases corresponding to very long S phase and G2/mitosis phases.

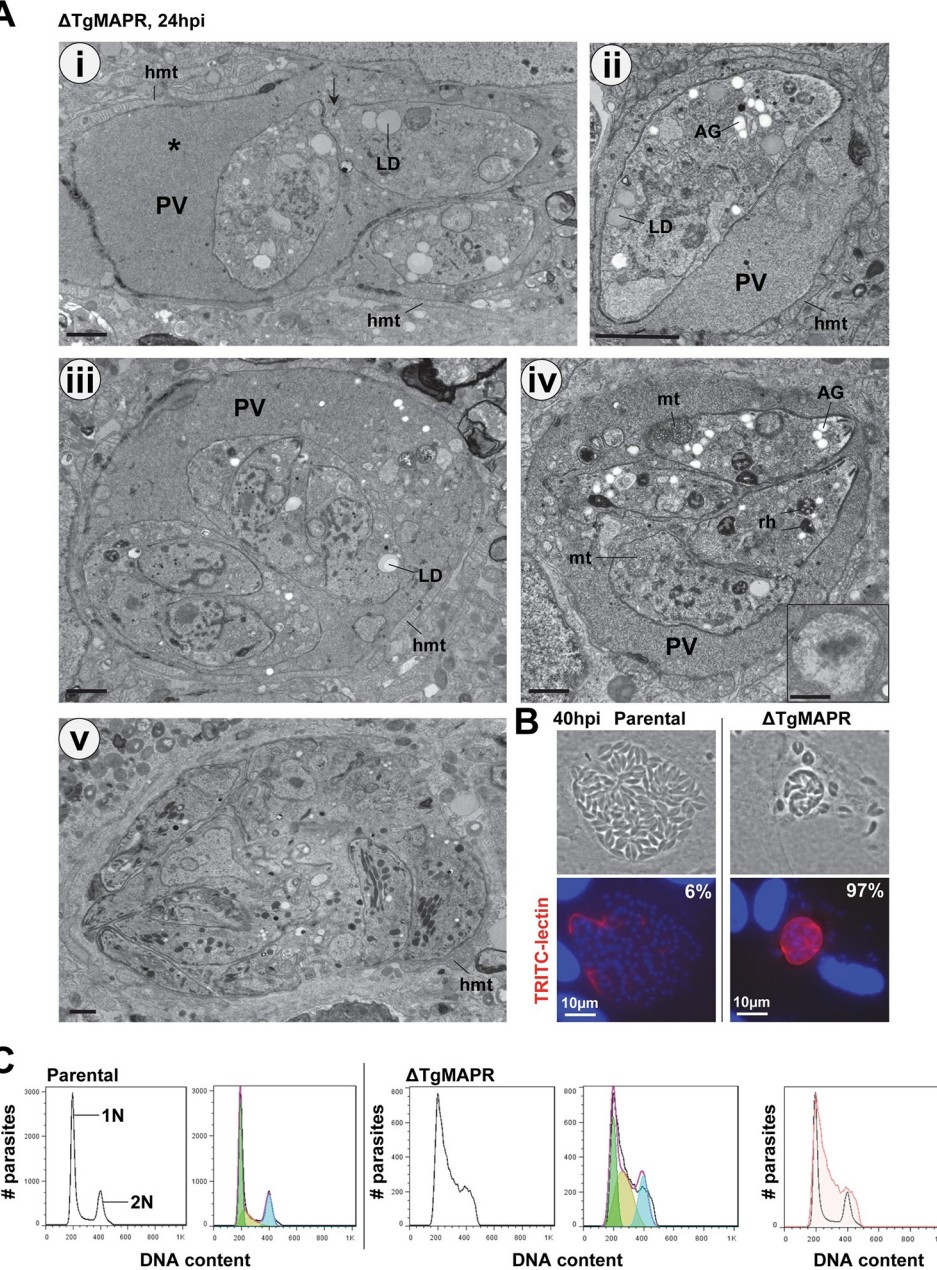

**Fig 4. Ultrastructure and cell cycle properties of ΔTgMAPR parasites.** (A) EM of intracellular ΔTgMAPR parasites 24 h p.i. Panels i to v illustrate parasites at different stages of replication in a PV matrix filled with granular material (asterisk in panel i with arrow showing an 'opening' at the basal end). The mutant contains many lipid droplets (LD) and amylopectin granules (AG), abnormal mitochondrion (mt) and rhoptries (rh). Inset in panel iv shows another example of an aberrant, enlarged mitochondrion. Host mitochondria (hmt) are recruited at the PV membrane. Scale bars, 250 nm. (B) Fluorescence microscopy on parental and ΔTgMAPR parasites using TRITC-lectin, showing a stronger signal on 97% mutant PV. n = 39–51 PV observed for each strain. (C) Flow cytometry-based DNA content measurements for cell cycle analysis of ΔTgMAPR and parental strains. The gating strategy is shown in S6A Fig. Fibroblasts were infected for 15 h before parasite isolation, ethanol-fixation and staining with the DNA-intercalating fluorescent dye propidium iodide to reveal DNA distribution patterns in the major phases of the cell cycle. The pseudo-color plot graphs show the population of parasites at 1N DNA phase (light green peak) and at 2N DNA phase (light blue peak), with very few WT parasites transitioning between 1N DNA and 2N DNA (PLVAC peak). In contrast, the khaki peak is more prominent in ΔTgMAPR$^{<7wks}$ parasites, indicating dysregulated cell cycle, with an abnormally long S phase. Representative images are shown from 4 independent assays.

## ΔTgMAPR parasites are able to overcome their growth defects overtime

Intriguingly, we observed that ΔTgMAPR parasites maintained *in vitro* for several weeks lysed their host cells more rapidly than initially, reflecting growth rate acceleration; this phenomenon appeared beyond 7 weeks of culture. We quantified this phenotype by plaque assays comparing the growth of 'young' clones ΔTgMAPR parasites (less than 7 weeks of culture; TgMAPR$^{<7wks}$) with culture-adapted mutant parasites (ΔTgMAPR$^{ad}$) and controls including parental and complemented parasites. To engineer the complemented strain, the *Tgmapr* gene with C-terminal HA tag has been integrated into the UPRT site of ΔTgMAPR parasites via homologous recombination; TgMAPR-HA expressed under the control of the tubulin promotor localized to the parasite mitochondrion as expected (S3D Fig). Lysed plaque area generated by TgMAPR$^{ad}$ were ~5-times larger than those from TgMAPR$^{<7wks}$; no statistical difference was observed between plaque area formed by ΔTgMAPR$^{ad}$, parental and complemented parasites (Fig 5A). We monitored the ability of ΔTgMAPR$^{ad}$ parasites to invade mammalian cells, and invasion assays show no difference in number of mutant parasites either attached or internalized into cells, compared to parental parasites (Fig 5B). Uracil incorporation assays showed no statistical difference in the replication rate between ΔTgMAPR$^{ad}$, parental and complemented parasites (Fig 5C). Time of egress upon A23187 induction revealed only a minor 1.1-fold delay for ΔTgMAPR$^{ad}$ compared to control parasites (Fig 5D). Finally, flow cytometry analyses conducted on ΔTgMAPR$^{ad}$ parasites revealed that their cycle had returned to normal, with the G1 interphase and S/G2/mitosis phases corresponding to $59.9 \pm 6.5\%$ and $38.1 \pm 5.3\%$, respectively (Figs 5E and S7B).

We performed RNA-Seq analysis on ΔTgMAPR$^{ad}$ versus ΔTgMAPR$^{<7wks}$ parasites to possibly identify genes with modified transcription levels, in relation to fitness recovery of the mutant. Eighteen genes were significantly differently expressed in ΔTgMAPR$^{ad}$ parasites, with 11 genes up-regulated and 7 genes down-regulated (S8 Fig). In searches of the *Toxoplasma* database (www.toxoDB.org) and SWISS-PROT database, the majority of the genes had either no known homologue or only an identifiable protein domain. Among the few genes for which a function could be attributed, at least one up-regulated gene (with a $\log_2$ fold change 4.4) putatively encodes a DNA primase, an enzyme involved in the DNA replication fork, which may correlate with the growth recovery of the cultured-adapted mutant.

These observations illustrate that ΔTgMAPR parasites have adapted to the loss of the *mapr* gene by some compensatory mechanisms allowing normal growth to resume, at least *in vitro*.

## ΔTgMAPR have reduced virulence regardless of culture adaptation

We next assessed the virulence of ΔTgMAPR$^{<7wks}$ and ΔTgMAPR$^{ad}$, compared to parental and complemented strains, in outbred mice. Survival curves of mice infected with control parasites (parental or complemented) show 100% mortality 11–12 days after infection (Fig 5F), similarly to WT parasites. By contrast, all mice infected with ΔTgMAPR died after 17–18 days regardless of their time in culture, indicating loss of virulence for the mutant. The similar reduction in virulence between ΔTgMAPR$^{<7wks}$ and ΔTgMAPR$^{ad}$ suggests that the *mapr* gene, although largely dispensable *in vitro*, contributes in part to the survival of *Toxoplasma* in mice.

## ΔTgMAPR$^{ad}$ partially recovered from membrane damage

Based on observations of restored fitness of ΔTgMAPR$^{ad}$ *in vitro* but reduced virulence in animals, we conducted EM analysis of ΔTgMAPR$^{ad}$ to examine to which extent the ΔTgMAPR$^{ad}$ parasites have recovered from their severe defects in endodyogeny (Fig 4A). We observed that the intravacuolar granular material was largely resorbed between ΔTgMAPR$^{ad}$ parasites for 65% of PV, although spare cellular debris and membrane whorls were visible in PV (Fig 6A,

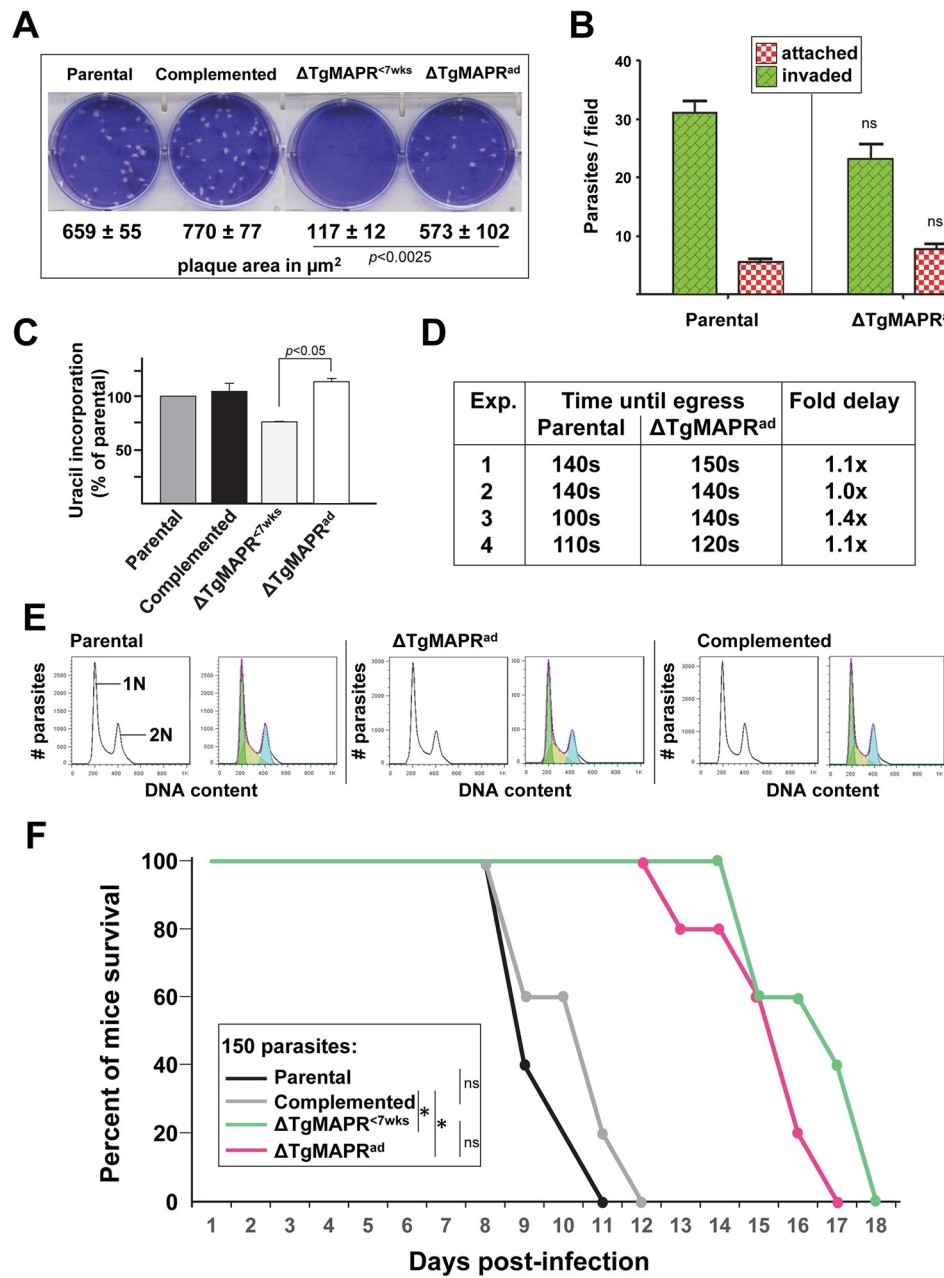

**Fig 5. Recovery of ΔTgMAPR adapted in culture and virulence.** See Fig 3 for legend description of the assays used. (A) Growth plaque assays for parental, complemented, ΔTgMAPR^{<7wks} and ΔTgMAPR^{ad} strains. (B) Red/green invasion assays. (C) Uracil incorporation assays. (D) Time-lapse microscopy egress assays. (E) Flow cytometry-based DNA content measurements for cell cycle analysis of ΔTgMAPR^{ad}, parental and complemented strains. The gating strategy is shown in S6B Fig. Representative images are shown from 4 independent assays, revealing culture-adapted mutant with S phase recovery. (F) Acute virulence assays of ΔTgMAPR before and after adaptation in a murine model. Swiss-Webster were intravenously inoculated with 150 parental, complemented, ΔTgMAPR^{<7wks} or ΔTgMAPR^{ad} parasites, with 8 mice for each strain to monitor mice mortality daily until Day18. *, $p = 0.0024$ (Log-rank Mantel-Cox test).

panel i). Some abnormalities in daughter cell budding still persisted in early stages of endodyogeny, with the formation of upside-down daughter cells with their apex abnormally facing the basal end of the mother cell (Fig 6A, panel i), predicting an unusual emergence of new

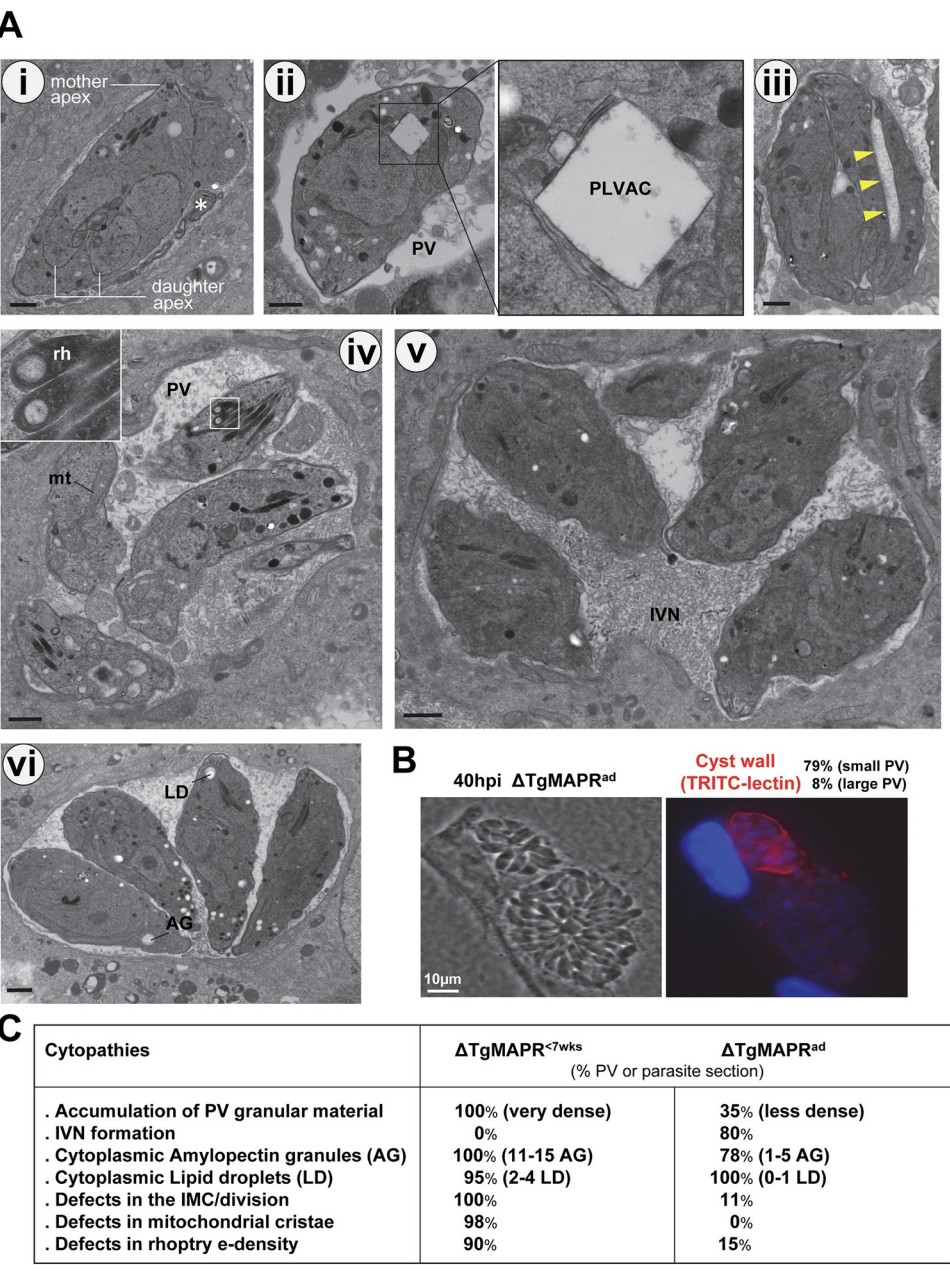

**Fig 6. Ultrastructure of ΔTgMAPR parasites adapted in culture.** (A) EM of intracellular ΔTgMAPR^ad parasites 24 h p.i. Panels i to vi illustrate parasites on the path of recovery with IVN formed, but still displaying some abnormalities in endodyogeny (arrowheads), accumulated cell debris in the vacuolar space (asterisk in panel i), amylopectin granules (AG), defects in rhoptry (rh) density and PLVAC localization. Scale bars, 250 nm. (B) Fluorescence microscopy on ΔTgMAPR^ad parasites using TRITC-lectin, showing a signal on small PV. (C) Table summarizing the cytopathies of ΔTgMAPR^ad and ΔTgMAPR^<7wks parasites observed on 72 to 97 PV or parasite sections.

parasites from the mother cell. Non dividing intravacuolar parasites (less than 10%) exhibited features of extracellular parasites. For example, the plant-like vacuolar compartment (PLVAC) that functions as an endo-lysosomal compartment is an enlarged single compartment, apically located in extracellular parasites; shortly after invasion, this organelle fragments into small vesicles that distribute throughout the parasite cytoplasm [57]. However, in intracellular ΔTgMAPR^ad parasites, PLVAC persisted as a single organelle and was aberrantly located at the basal

end, suggesting abnormal remodeling of PLVAC following invasion (Fig 6A, panel ii). Late stage of budding, which is reflected by the incorporation of the mother's plasma membrane into each emerging daughter synchronously, showed abnormal cleavage separating the daughter cells of ΔTgMAPR[ad] parasite (Fig 6A, panel iii, arrowheads). Most PV of ΔTgMAPR[ad] parasites contained several dividing parasites with individualized progenies (Fig 6A, panels iv to vi), a sign of replication recovery. An IVN was clearly defined in the PV lumen of 80% ΔTgMAPR[ad] parasites (Fig 7A, panel v). The number of lipid droplets and the mitochondrial morphology returned to normal, but ΔTgMAPR[ad] parasites still contained anomalously electron-dense rhoptries (Fig 6A, panel iv) and many amylopectin granules (Fig 6A, panel vi). Lectin fluorescence binding assays showed a lectin-positive signal for 79% of PV (small size with 1 to 16 parasites) vs. 8% of large PV with >16 parasites (Fig 6B). A table in Fig 6C summarizes the ultrastructural cytopathies of ΔTgMAPR[<7wks] and ΔTgMAPR[ad] parasites.

Overall, these morphological observations are largely compatible with fitness restoration of ΔTgMAPR[ad] parasites in cultured cells but their residual cytopathies, either irreversible or

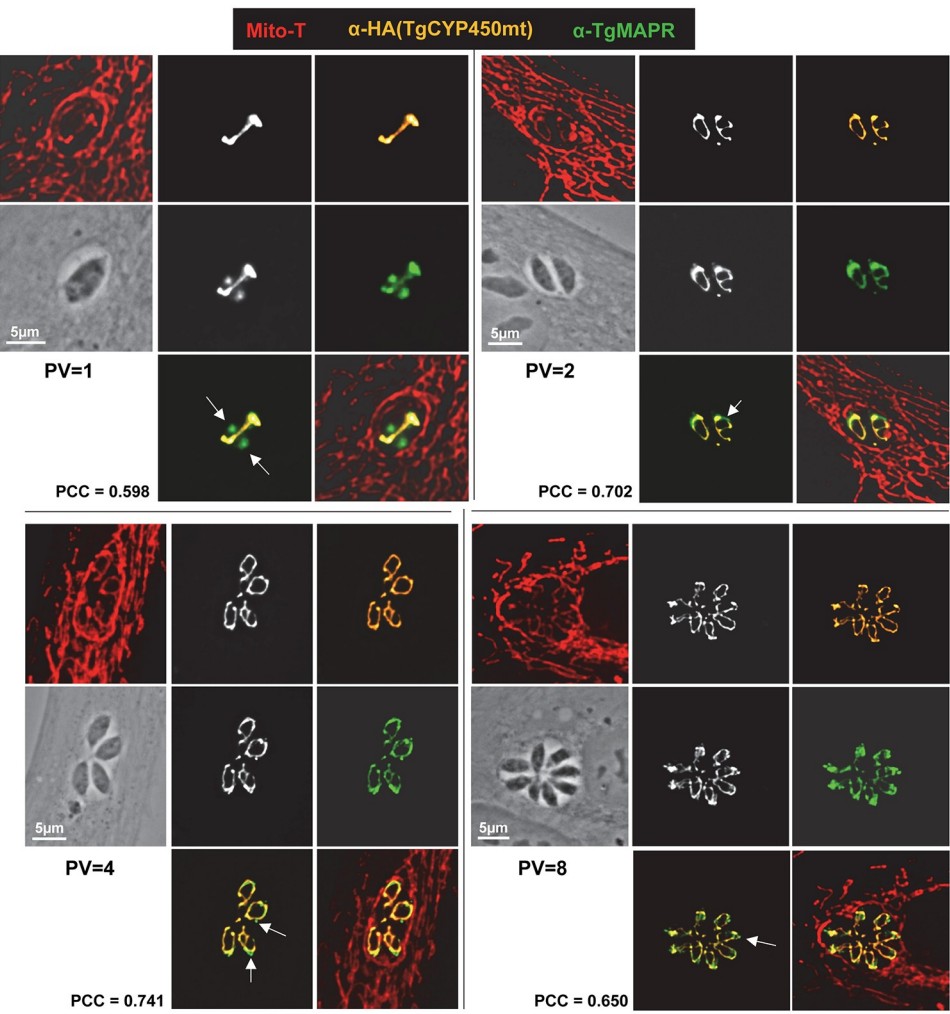

**Fig 7. Distribution of TgMAPR in *Toxoplasma* relative to TgCYP450mt.** Double IFA on TgCYP450-HA-expressing *Toxoplasma* at different stages of replication using anti-HA and anti-TgMAPR antibodies showing colocalization in the parasite mitochondrion staining with Mito-Tracker. Values of PCC are shown for the HA and TgMAPR signals. Arrows on the merged images point to areas exclusively containing TgMAPR (green).

inherent to the permanent *mapr* loss, could explain weakened virulence under harsh conditions, such as external pressure from the host.

## TgMAPR and TgCYP450mt colocalize in the mitochondrion

We next investigated to which extent TgCYP450mt codistributes with TgMAPR in the parasite mitochondrion. Double IFA on TgCYP450-HA-expressing parasites using anti-HA and anti-TgMAPR antibodies illustrated a significant colocalization between the 2 signals in the mitochondrion identifiable by Mito-Tracker in parasites in small and large PV (Fig 7). Interestingly, the signal for TgMAPR seems to extend beyond the signal for Mito-Tracker and TgCYP450mt (Fig 7; arrows).

## TgCYP450mt and TgMAPR form dynamic in situ interaction in the mitochondrion

To assess whether TgCYP450mt and TgMAPR interact with each other in the parasite mitochondrion, we performed in situ proximity ligation assays (PLA) in TgCYP450-HA-expressing parasites. No PLA signals were detected when both primary antibodies (anti-HA or anti-TgMAPR) were omitted or when only one of the primary antibodies was used (Fig 8A). When the primary antibodies anti-HA and anti-TgMAPR were incubated for 2 h at 37°C on fixed parasites, a robust PLA signal was detected within the parasite, suggestive of the presence of TgCYP450mt:TgMAPR complexes (Fig 8B). PLA analysis on small and large PV revealed in all instances bright fluorescent dots within the parasite, sometimes aligned in a configuration reminiscent of the shape of the mitochondrion. The mitochondrial-localized HSP70 strongly colocalizes with TgCYP450mt (Fig 1B) and TgMAPR (S5E–S5F Fig). An additional control consisted of PLA in the presence of anti-HA (in TgCYP450-HA-expressing parasites or in TgMAPR-HA-expressing parasites) and anti-HSP70, and data showed no fluorescent signals in these parasites, indicating no interaction between HSP70 and TgCYP450mt or TgMAPR (S9 Fig). These observations indicate that TgCYP450mt and TgMAPR are proximal to each other, suggesting interactions of TgCYP450mt and TgMAPR in the parasite mitochondrion.

## ΔTgMAPR and iΔTgCYP450mt *Toxoplasma* have reduced levels of steroids compared to WT parasites

The close proximity of TgCYP450mt and TgMAPR in the parasite mitochondrion and the shared sequence identity (up to 36%) of TgCYP450mt with human mitochondrial CYP450, prompted us to examine the contribution of TgCYP450mt and TgMAPR to steroid production by conducting lipidomic analysis on iΔTgCYP450mt (2 days of ATc treatment), ΔTgMAPR$^{<7\text{wks}}$ (pre-adapted strain) and WT parasites. We selected six steroids (progesterone, corticosterone, aldosterone, DHEA, cortisone and cortisol) that were detected in our LC-MS analysis (Table 1) to compare their levels in the 3 parasite populations. Based on raw intensity a.u. data normalized by protein concentration, all six steroids were detected in lower amounts in the two mutant populations compared to WT, with iΔTgCYP450mt parasites showing a more dramatic reduction, corresponding to 10- to 25-fold (Fig 9). These data point to a biosynthetic pathway for specific steroids in *Toxoplasma*, involving TgCYP450mt activity, possibly enhanced by TgMAPR. However, caution must be exerted in the interpretation of this finding as WT, ΔTgMAPR and iΔTgCYP450mt parasites have different growth rates, and thus upon collection at different time points, they were at different stages of their lytic cycle (intracellular in small PV, large PV, or extracellular), and it is unknown whether the steroid production is continuous through the parasite life cycle. Furthermore, the mitochondrion in

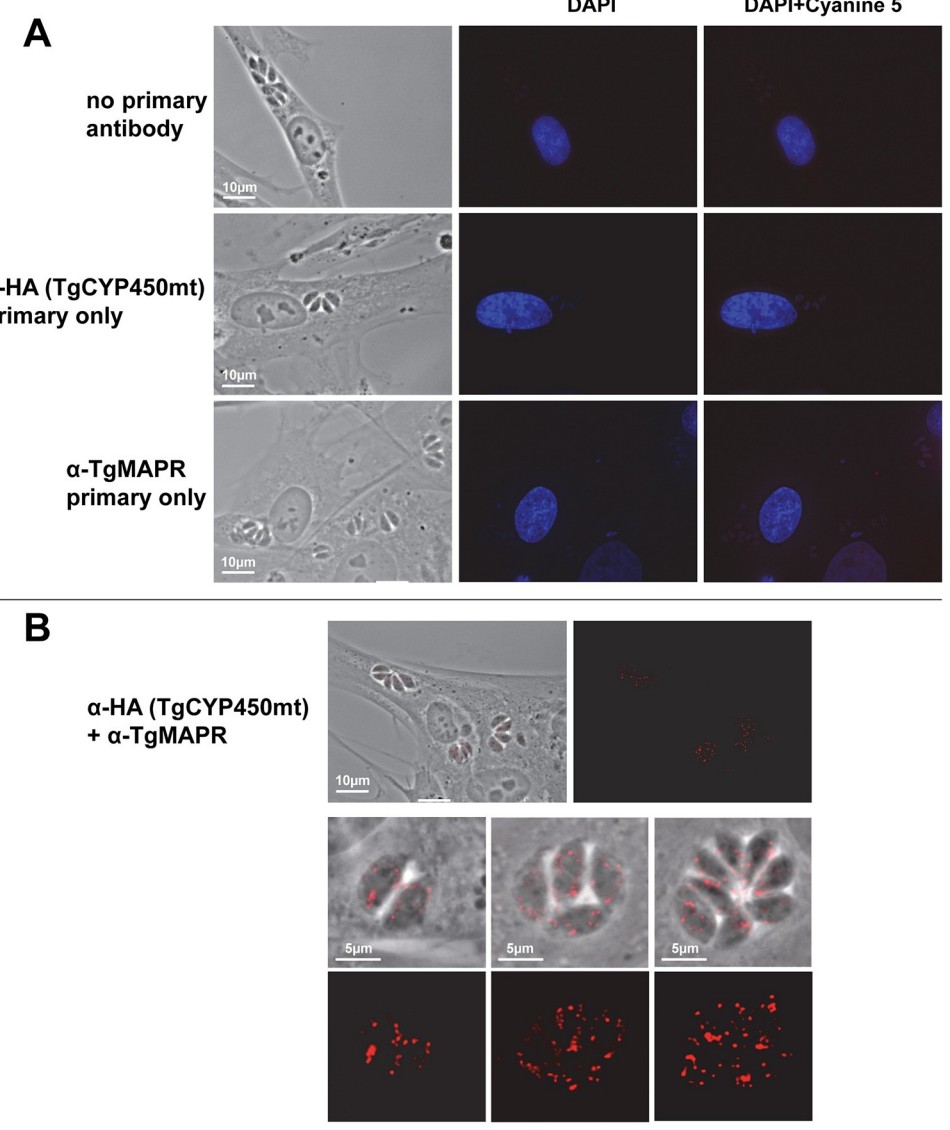

**Fig 8. In situ interaction between TgCYP450mt and TgMAPR.** (A) Fluorescence microscopy of TgCYP450-HA-expressing *Toxoplasma* using a proximity ligation assay (PLA) showing punctate fluorescent signal in red upon TgCYP450mt:TgMAPR interaction. No signal for TgCYP450mt:TgMAPR interaction was observed when both or either one of the primary protein antibodies was omitted during the PLA process. Interaction of both proteins is visible by PLA only when both primary antibodies for TgCYP450mt (anti-HA) and TgMAPR were included. Nuclei staining by DAPI. (B) PLA on various PV sizes of TgCYP450-HA-expressing *Toxoplasma* using anti-HA and anti-TgMAPR antibodies showing specific signal delineating mitochondrial profiles.

TgMAPR parasites has an aberrant morphology (although reversible), making it plausible that the reduced steroid levels may be due to impaired mitochondrial functions that indirectly impact steroidogenesis.

## Discussion

Steroid hormones are modulators of many cellular functions essential for cell growth, homeostasis and energy production by regulating the transcription of a large number of genes through interaction with specific receptors located at the nucleus, plasma membrane or

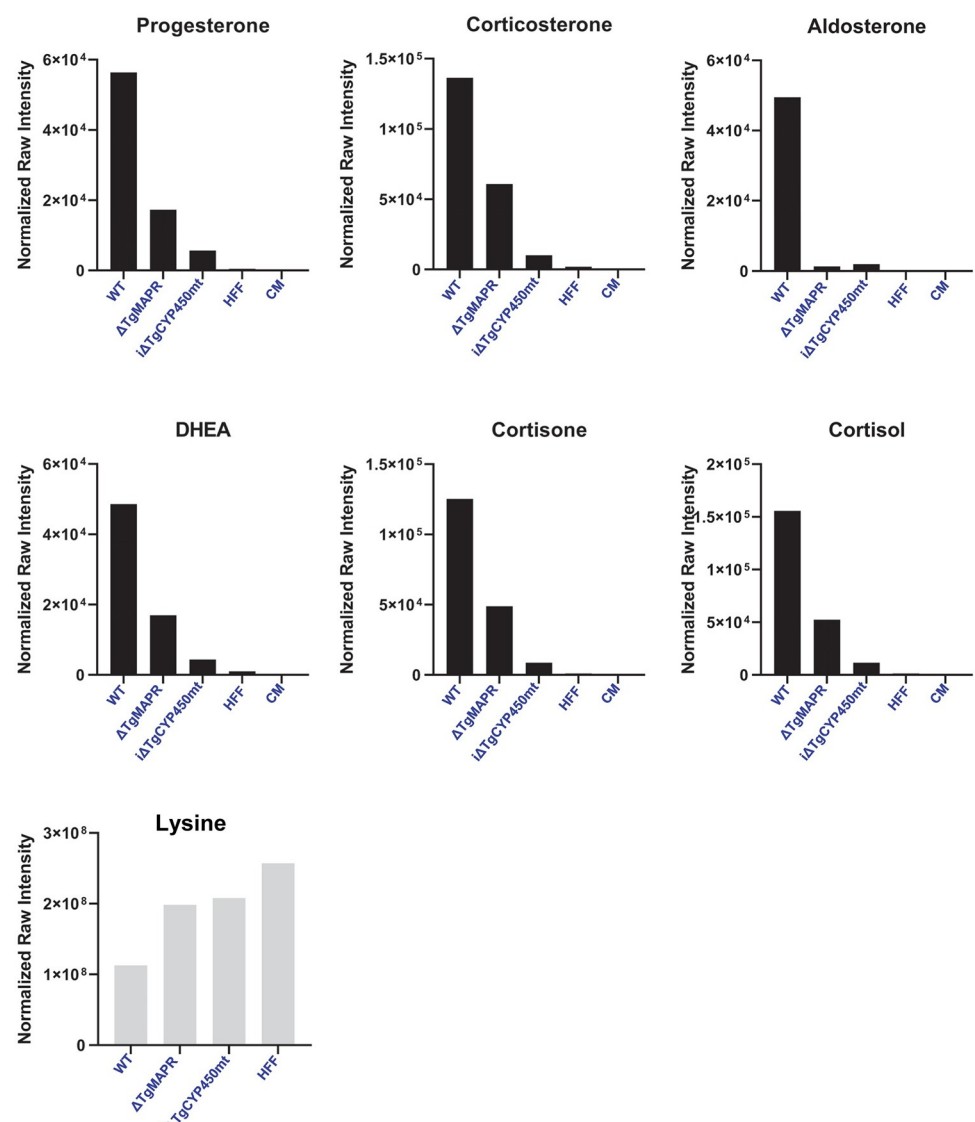

**Fig 9. Comparative levels of steroids in ΔTgMAPR and iΔTgCYP450mt to WT *Toxoplasma*.** LC-MS for comparative analysis of the indicated steroids in purified extracellular ΔTgMAPR, iΔTgCYP450mt and WT parasites, showing reduced levels of steroids in the two mutants compared to WT. Controls for extra-parasitic metabolite contamination included HFF and culture media (CM). DL-Lysine measurement was used as a verification of our normalization based on protein concentration for the four cellular preparations. Data are means of two samples and expressed as raw intensity (normalized to protein concentration) of each steroid standard and Lysine.

cytosol. Our study uncovers that steroid metabolism may exist in the mitochondrion of *Toxoplasma gondii*, likely involving TgCYP450mt and TgMAPR based on lower levels of steroids detected in parasites with reduced expression of these proteins. Based on steroidogenesis in other organisms, our hypothetical model for cholesterol biotransformation to steroids in the parasite mitochondrion would involve TgMAPR dimerization to favor interaction with TgCYP450mt for stabilization [37]. A potential source of cholesterol may be cholesteryl esters stored in lipid droplets that would need to be de-esterified by a cholesterol esterase (uncharacterized so far). Free cholesterol in the cytosol may then be transported to the outer mitochondrial membrane after binding to the lipid transfer protein TgHAD-2SCP-2 [58], by analogy to mammalian SCP-2 that mediates cholesterol transport to mitochondria [59]. Import of

cholesterol within the mitochondrion may involve a parasite steroidogenic acute regulatory (StAR) homolog to StAR domain-containing protein (STARD), one of the main proteins of the transduceosome that spans the two mitochondrial membranes and is required for cholesterol translocation [60]. Both STARD1 and STARD3/MLN64 bind cholesterol and move this lipid across membranes. The gene TGME49_236660, in the *Toxoplasma* database codes for a close homolog to STARD3 that functions in cholesterol transport to the mitochondria and to the cell membrane [61,62]. The TGME49_236660 sequence (predicted MWt: 150.3-kDa) harbors a StAR-related domain at the C-terminus (from 1211-1380aa). Within the parasite mitochondrion, cholesterol can be converted to pregnenolone and then to steroid hormones. The enzymatic steps involved in the production of steroids from cholesterol in mitochondria in Vertebrates are illustrated in S10 Fig, in which the mitochondrial enzymes with functions compatible to TgCYP450mt have been highlighted.

Members of the protozoan family Sarcocystidae encode one *MAPR* gene homolog and this study reveals that MAPR in *Toxoplasma* interacts with a mitochondrial CYP450 enzyme involved in cholesterol metabolization. The genome of Sarcocystidae does not contain genes encoding microsomal CYP450. The evolutionary origin of eukaryotic MAPR proteins and mitochondrial CYP450 is still a matter of debate. A phylogenetic reconstruction proposes that the original eukaryotic MAPR proteins are related to prokaryotic cytochrome-b5 domain proteins, with similarities in the MAPR-like folded architecture, heme-binding orientation and MAPR-like heme-interacting tyrosines [63]. MAPR proteins are multifunctional, with some functions present in ancient eukaryotes/yeast related to activation of CYP450 (e.g., sterol biosynthesis, drug metabolism, iron homeostasis) and other specialized functions acquired later in eukaryotic lineages and metazoan development (e.g., cell cycle regulation, membrane trafficking, axon guidance, fertility) [64]. Concerning the origin of mitochondrial steroidogenic CYP450 enzymes, a phylogenetic study refutes their acquisition from an aerobic prokaryote (mitochondrial endosymbiont) because CYP450 in bacteria and archaea are soluble while mitochondrial CYP450 of eukaryotic organisms are membrane-bound [31]. Instead, this study proposes that a microsomal CYP450 is the ancestor of animal mitochondrial P450, following the conversion of the ER-targeting sequence of microsomal CYP450 to a mitochondria-targeting sequence and adaptation to receive electrons from ferredoxin/adrenodoxin. Many studies claim that the appearance of CYP450 in mitochondria of eukaryotes has occurred after animal lineage diverged from fungi and plants [65,66]; however, helminths contain nuclear hormone receptors, steroidogenic enzymes and steroid hormones (e.g., estrogens, testosterone) for the development and reproduction of several species [67,68]. Nevertheless, whether the emergence of a biosynthetic pathway for steroid-like molecules involving a cytochrome-b5 domain-containing protein has occurred first in bacteria or in eukarya, some bioinformatic evidence [39] and our work point that some early protists have a MAPR homolog that may be involved in steroidogenic regulation. Furthermore, another study showed that *Trypanosoma cruzi* can metabolize dehydroepiandrosterone (DHEA) to precursors of testosterone such as androstenedione and androstenediol, and can convert exogenous androstenedione to testosterone, 17β-estradiol and estrone [69].

We showed that *T. gondii* contains at least 14 different steroid molecules and expresses TgCYP450mt that shares identity and functionality with mammalian steroidogenic enzymes. Genetic ablation of the TgCYP450 gene is not tolerated by *Toxoplasma*, which leads to the assumption that steroid production by the parasite is associated with essential functions for virulence. Detection on TgCYP450mt and TgMAPR in the mitochondrion in close proximity points to an interrelationship between the two proteins by analogy to other organisms, with TgMAPR acting as a binding partner to stabilize/activate TgCYP450mt, with functions related to steroid metabolism.

ΔTgMAPR have pleotropic phenotypes. First, they suffer from replication and growth defects, which implicates TgMAPR in *Toxoplasma* proliferation, analogously to mitochondrial PGMRC1 in cancer cells that promotes tumor growth and metastasis [70]. Second, TgMAPR-deficient parasites show dysregulation in their cell cycle, reflected by an abnormally high proportion of mutant parasites in the S-phase of the cell cycle and few parasites that progress to cytokinesis; this suggests a contribution of TgMAPR to progression into or through G1. Third, TgMAPR-deficient parasites leak their content into the PV lumen, indicating that TgMAPR may preserve membrane permeability. Fourth, the mitochondrion of these mutants exhibits abnormal cristae, suggesting that TgMAPR regulate mitochondrial membrane maintenance, similar to Dap1 that is required for mitochondrial stability in yeast [71]. It would be interesting to examine whether there is an association of the metabolic effects of TgMAPR with steroidogenesis, which would ascribe a protective role of steroids in cell cycle regulation and survival of *Toxoplasma*.

In addition to association with steroidogenic-associated CYP450 enzymes [reviewed 18], mammalian PGRMC1 have other binding partners. When localized to mitochondria (usually the outer membrane), PGRMC proteins interact with ferrochelatase [72], the terminal enzyme in the heme biosynthetic pathway, converting protoporphyrin IX into heme. *Toxoplasma* scavenges free heme from the host, but also has a heme biosynthetic pathway [73], with a ferrochelatase enzyme predicted to localize to the mitochondrion and be essential for parasite growth and acute virulence [74,75]. Thus, in addition to its chaperoning role in steroid production, TgMAPR may function as a heme chaperone for the parasite ferrochelatase.

Deletion of the *cyp450mt* gene is not tolerated by *Toxoplasma*, which could be surprising in view that steroidogenesis is not required for the survival of a eukaryotic cell. Possible interpretations would be that TgCYP450mt is a monofunctional enzyme acting upstream of a pathway that produces essential metabolites or is a multifunctional enzyme operational in different pathways, with a broad impact on parasite physiology. If TgCYP450mt has several substrates, our data point to sterol/steroid as one of them, but it is difficult to predict, at this stage, the nature of additional substrates. The physiological relevance of steroids for *Toxoplasma* remains to be investigated. During the evolution of multicellular organisms, hormone steroids have acquired specialized functions (e.g., reproduction, inflammation, salt balance). It is possible that steroids for an ancient unicellular organism, may have primitive yet important roles in defense mechanisms or survival.

Steroid metabolism could be advantageous for *Toxoplasma*. In non-professional steroidogenic cells, steroids can exert functions through three regulatory mechanisms: intracrine (without being secreted from the cell of origin), autocrine (same as intracrine but in response to an external stimulus), or paracrine (being secreted to modulate neighboring cells). Hormone steroids can activate signaling pathways in the cytosol and control gene transcription in the nucleus and the mitochondrion. Genomic analysis reveals that *Toxoplasma* contains zinc finger HIT domain-containing proteins that bind to nuclear hormone receptors [25]. If present, it would suggest that steroids produced by *Toxoplasma* may bind to parasite steroid receptors to control their own functions, in a cell-autonomous intracrine fashion. Alternatively, *Toxoplasma* may generate and secrete steroids into the host cytosol to modulate host cellular metabolism and immune function. *Toxoplasma* is proficient in evading immune responses including long lived inflammatory response during chronic and acute infection of any hosts (virtually all warm-blooded animals). This suggests universal mechanisms that require an immune modulating currency that most mammals would respond to, and is easily transportable outside of the parasite. One would have extreme difficulty finding an immune modulating molecule more universal, and more disseminatable, than steroids. Many steroids produced by *Toxoplasma* have anti-inflammatory properties. For example, pregnenolone and its abundant

metabolite 17-hydroxypregnenolone promotes ubiquitination and degradation of the TLR2/4 adaptor protein TIRAP and TLR2 in macrophages and brain cells, and suppresses the secretion of TNFα and IL-6 mediated through TLR2 and TLR4 signaling [76]. Of the other abundant steroids in *Toxoplasma*, DHEA affects cytokine secretion by T lymphocytes [77] and corticosterone impairs MHC class I antigen presentation by dendritic cells [78,79], involved in anti-*T. gondii* immunity [80]. Progesterone has an immunosuppressive effect on the innate immune responses and attenuates inflammatory responses in the brain [81], the site of *Toxoplasma* encystation and chronic infection. Finally, it remains to be determined if *Toxoplasma* can salvage steroids from the host either to utilize them for its own gene regulation or for catabolism, such as observed for intracellular bacteria, *Mycobacterium* and *Rhodoccus* that access cholesterol and steroids (e.g., androstenedione) as carbon sources from their host and whose pathogenicity correlates with these sterol/steroid catabolic activities [82–84]. Future extensive studies will include the functional characterization of the enzymatic step(s) mediated by TgCYP450mt, identification of other potential binding partners of TgMAPR and analysis of the protective roles of parasite steroids and receptors in the pathogenicity of *Toxoplasma*.

## Materials and methods

### Ethics statement

This study was performed in compliance with the Public Health Service Policy on Humane Care and Use of Laboratory Animals and Association for the Assessment and Accreditation of Laboratory Animal Care guidelines. The animal protocol (# MO20H368 –Coppens as the Principal Investigator) was approved by Johns Hopkins University's Institutional Animal Care and Use Committee on 02/01/2021.

### Mice and in vivo virulence assays

5 weeks-old female Swiss-Webster mice were purchased from The Jackson Laboratory (Bar Harbor, ME). Groups of mice (6 or 8 per group) were infected intraperitoneally or intravenously with *Toxoplasma* mutant or control (parental or complemented) strains. The survival of mice after infection was monitored daily. To assess the virulence of iΔTgCYP450 tachyzoites, cultured parasites were cultivated *in vitro* for 24 h in the presence of 1 μg of ATc/ml prior to injection in mice; to maintain low expression of TgCYP450, drinking water was supplemented with 0.2 mg of ATc/ml. Eighteen- or 22-days post-infection, the mice were euthanized by carbon dioxide inhalation.

### Reagents and antibodies

All chemicals were obtained from Sigma (St Louis, MO) or Fisher (Waltham, MA) unless otherwise stated. [5,6-$^3$H]uracil was purchased from Dupont NEN Corporation (Boston, MA). Lipid standards are from Avanti Polar Lipids (Birmingham, AL). Crystals of hemin were purchased from Frontier Scientific, Inc. (Logan, UT). Primary antibodies include mouse anti-TgMAPR (this study); mouse and rabbit-anti-SAG1, mouse anti-GRA3, mouse anti-GRA1, mouse anti-BAG1 all provided by J.F. Dubremetz (University of Montpellier, France); rabbit anti-GRA7 [3]; rabbit anti-HSP70 provided by D. Soldati (Université de Genève, Switzerland) [32]; mouse anti-TgABCG [5]; mouse anti-HA (BioLegend, San Diego, CA); rabbit or mouse anti-HA (Babco, Richmond, CA or Bethyl Laboratories Montgomery, TX); rabbit anti-DsRed (Takara Bio, Mountain View, CA). Secondary antibodies used for immunofluorescence conjugated to Alexa488 or Alexa594 were purchased from Invitrogen (Carlsbad, CA). Secondary

antibodies used for immunoblots are horseradish peroxidase-conjugated goat anti-mouse and donkey anti-rabbit IgG (GE Healthcare, UK).

## Cell lines, parasites and culture conditions

Human foreskin fibroblasts (HFF), VERO cells and HeLa cells obtained from the American Type Culture Collection (Manassas, VA), and immortalized HFF (hTERT) received from S.N. J. Moreno (University of Georgia), were all grown as monolayers and cultivated in α-minimum essential medium (αMEM) supplemented with 10% fetal bovine serum (FBS), 2 mM glutamine and penicillin/streptomycin (100 units/ml per 100 μg/ml), and maintained at 37˚C in 5% $CO_2$ unless specified otherwise. For some assays, cells were incubated in phosphate-free DMEM (Gibco, Gaithersburg, MD). The tachyzoites from the RH strain (Type I lineage) used in this study were propagated *in vitro* by serial passage in monolayers of HFF as described [85].

## Sequence analysis

Nucleotide and amino acid sequences were searched against the *T. gondii* database (GT1 or ME49 strain; www.toxoDB.org) and the NCBI database using the BLAST algorithm [86]. Multiple sequence alignments were created using ClustalW, and the resulting similarities were then visualized by subjecting the alignment to Boxshade (www.ch.embnet.org). Percent identity and similarity were calculated using standard tools for sequence analysis from NCBI (ncbi. nlm.nih.gov).

## Cloning of full-length cDNA encoding TgMAPR and stable transfection of TgMAPR-HA in *Toxoplasma*

BLAST searches in the *Toxoplasma* database (www.toxoDB.org) identified a membrane-associated progesterone receptor homolog (TgMAPR, TGME49_276990). mRNA of TgMAPR was isolated from parasites freshly lysed out from mammalian cells and transcribed into first-strand cDNA. For stable transfection of TgMAPR-HA, the 289-bp TgMAPR coding sequence was amplified using primers ProgR-F1 and ProgR-HA-R1 (S1 Table), and subsequently cloned into plasmid pYFP (BglII/AvrII) [87] allowing the expression of C-terminally HA-tagged TgMAPR-HA under the control of the tubulin promoter. Extracellular parasites ($10^7$) were transfected with the TgMAPR-HA construct by electroporation and stable parasites selected using 20 μM chloramphenicol and cloned by limiting dilution; correct clones were identified based on HA signal using anti-HA antibodies.

## Recombinant expression of TgMAPR in *E. coli*

N-terminally 6xHis-tagged TgMAPR (6xHis-TgMAPR) was cloned into pQE30 plasmid using primers ProgR-F3 and His-ProgR-R5 (BamHI/HindIII) and expressed in the M15 *E. coli* strain. Cultures were grown at 37˚C until an $OD_{600}$ of ~ 0.5 and induced with 1 mM IPTG for 5 h at 25˚C. Cells were harvested by centrifugation and lysed in lysis buffer (50 mM $NaH_2PO_4$, 300 mM NaCl, 10 mM Imidazole, pH 8.0) containing 1 mg/ml lysozyme and 1% N-Lauroyl sarcosine followed by 5 cycles of sonication. Cell debris was cleared by centrifugation and the recombinant protein was purified from the resulting supernatant under denaturing conditions on $Ni^{2+}$-NTA resin, according to the Qiagen protocol. The recombinant protein was analyzed by SDS-PAGE and by Coomassie staining or anti-6xHis western blot, and protein concentration was determined by Bradford assay [88]. After purification, the peptide was refolded by diluting the sample 10-times in the refolding buffer (50 mM Tris/HCl pH 7.5, 1 mM EDTA, 1

M L-arginine, 1 mM reduced form of glutathione, 0.8 mM of oxidized form of glutathione) at 4°C overnight. After dialysis against PBS, the protein was injected into mice for polyclonal anti-TgMAPR antibody generation (Covance Research Products Inc., Denver, PA). Recombinant 6xHis-TgMAPR peptide was also used for heme detection (see below).

## Western blotting

*Toxoplasma* tachyzoites were isolated by forced passage through a 25-gauge needle, filtration using 3μm membranes, and centrifugation at 400 *g* for 10 min in Hanks' balanced salt solution (HBSS) supplemented with 1 mM ethylene glycol-bis (β-aminoethyl ether, tetraacetic acid (EGTA) and 10 mM HEPES. Purified parasites resuspended at $10^8$ cells per ml in calcium/ magnesium-free PBS containing 1 mM EGTA, were subjected to three freeze-thaw cycles in the presence of protease inhibitors at the following concentrations: 10 μg/ml Nap-tosyl-L-lysine chloromethyl ketone (TLCK), 10 μg/ml p-aminophenylmethyl sulfonyl fluoride (a-PMSF), 1 μg/ml leupeptin, 10 μg/ml E-64. Freeze-thaw lysates were separated into a soluble fraction (supernatant) and membrane-associated fraction (pellet) by centrifugation at 100,000x*g* for 2 h in a TL-100.3 rotor using a Beckman Optima TL table-top ultracentrifuge. To determine the stability of proteins associated with membranes, aliquots of the membrane pellets were treated with 1% Triton X-100, 0.1 M $Na_2CO_3$ (pH 11.5) or 50 mM Tris-HCl for 30 min, followed by centrifugation at 100,000x*g*. Supernatants were concentrated by trichloroacetic acid precipitation and analyzed by SDS-PAGE followed by western blotting to detect TgMAPR at 26-kDa (anti-MAPR antibodies at 1:2,000), TgABCG at 96-kDa (anti-TgABCG$_{96}$ antibodies at 1:500) and GRA1 at 24-kDa (anti-GRA1 antibody at 1:1,000).

## Heme-binding properties of recombinant TgMAPR

A few crystals of hemin were suspended in 100 μl of 0.1 M NaOH, mixed thoroughly, and neutralized with 900 μl NaPi buffer (50 mM pH 7.2). Next, the hemin solution was mixed with purified 6xHis-TgPMAPR (ratio 1.5:1) and gently mixed at 4°C overnight. Excess free hemin was removed by nickel chromatography and reconstituted protein eluted with 20 mM imidazole. Heme-binding properties of TgMAPR were demonstrated by a scan of absorbance from 350–700 nm before and after oxidization with KFeCN. Alternatively, reconstituted protein was run on SDS-PAGE, subjected to in-gel heme staining with ECL solution and visualization by exposure to X-ray film [42]. Briefly, protein suspensions were incubated for 15 min at 42°C in 124 mM Tris (pH 7), 20% glycerol, 4.6% SDS. After electrophoresis on SDS-PAGE and transfer to nitrocellulose filters, the ECL detection reagents were used according to the manufacturer's recommendations, with the detection solution added directly to the blot for 1 to 10 min. Under the denaturing electrophoretic conditions, only proteins containing covalently bound prosthetic group heme retain their heme and are identified on blots by virtue of their heme peroxidase activity. A commercial peptide of hemoglobin containing heme-binding domain was used as a positive control.

## Transient transfection of TgMAPR-HA in HeLa cells

TgMAPR-HA was cloned into the mammalian expression plasmid pcDNA3.1⁻ (EcoRI/HindIII) using primers ProgR-F2/ProgR-HA-R2 allowing expression of a C-terminally HA-tagged protein under the control of the p*CMV* promoter. The construct was transfected into HeLa cells using lipofectamine, according to manufacturer's instructions, and the subcellular protein localization was visualized by immunofluorescence using mouse-anti-HA antibodies.

## Genetic disruption of TgMAPR and complementation of the Δtgmapr strain

To create the Δtgmapr strain, 1.5-kb 5'- and 3'-UTR fragments of TgMAPR were amplified from genomic DNA (primer pairs ProgR-5UTR-F1/R1 and ProgR-3UTR-F1/R1) and cloned into plasmid pDHFR-TS(2854) [87] using ApaI/NheI and SpeI/NotI restriction sites, respectively. Extracellular tachyzoites of the *Δku80* strain (~$10^7$) were transfected with ~50 μg of ApaI-linearized DNA and transgenic parasites were selected using pyrimethamine. Stable parasites were cloned and screened by PCR for homologous crossovers (primer pairs ProgR-5KOV-F1/DHFR-R1 and DHFR-F1/ProgR-3KOV-R1) and the absence of the MAPR gene locus (ProgR-ISP-F1/ ProgR-PD-3UTR-R1). We complemented the Δtgmapr strain by knock-in of TgMAPR-HA at the TgUPRT locus. The sequence of the tubulin promoter followed by TgMAPR-HA was amplified from construct TgMAPR-HA-pYFP (mentioned above) using primers PtubProgR-comp-F1/PtubProgR-comp-HA-R1 and subcloned into plasmid pUPRT (ApaI/HindIII) [87]. Δtgmapr were transfected with ~50μg of PciI-linearized construct and stable clonal parasites were obtained by selection with 20 μM 5-fluorodeoxyuridine (FUDR). Presence and proper expression of the complementation construct was verified by immunofluorescence using mouse-anti-HA antibodies.

## Cloning of plasmid constructs and transformation in *Schizosaccharomyces pombe*

TgMAPR-myc was cloned into the *S. pombe* expression plasmid pSLF101 provided by Peter Espenshade at the BamHI/PstI site using primers ProgR-F3/ProgR-myc-R3. TgMAPR-myc, SpDap1-3xHA (positive control), and pSLF101 (empty plasmid control) were transformed into a Dap1-deficient yeast mutant (ΔSpDap1; PEY901) using the lithium acetate method. Briefly, PEY901 cells were cultured in YES media at 30°C to an $OD_{600}$ of ~ 0.5–1.0 (20 ml/transformation). 10 OD of cells were collected and sequentially washing in sterile water and 1x TE/LiOAc made from 10x filter sterilized stocks (10x TE: 0.1 M Tris-HCl pH 7.5 and 0.01 M EDTA; 10x LiOAc, 1 M LiOAc pH 7.5). Cells were resuspended in 100 μl of TE/LiOAc per transformation, supplemented with 2 μl of sheared herring sperm DNA (10 mg/ml from Research Genetics; Huntsville, AL) and 1 μg of transforming DNA and incubated at room temperature for 10 min. Next, 260 μl of 40% PEG/TE/LiOAc (made up fresh from 50% PEG 4000, 10x TE and 10x LiOAc) were added and the mixture was incubated at room temperature with rotation for 30–60 min. Upon addition of 43 μl of DMSO, cells were vortexed and subjected to heat shock at 42°C for 5 min prior to plating on selective plates (EMM (-) Leu). Successful transformation was verified by Western blot on total yeast extract. Pellets of 2 OD of cells were resuspended in 100 μl of water, supplemented with 17 μl of 1.85 M NaOH/7.4% (v/v) βME and incubated on ice for 10 min. Protein was precipitated using the TCA/acetone method and analyzed by Western blot using mouse-anti-myc (1:1000) or rabbit-anti-HA (1:1000) antibodies.

## Hypoxia tolerance and lipid analysis of *ΔSpDap1* complemented with TgMAPR-myc

The lipid composition of transformed yeast strains was analyzed by gas chromatography (GC). 10 OD of yeast cells were resuspended in 1 ml methanol in a 20x150 mm glass disposable culture tube with a loose metal cap on top. An additional 8 ml of methanol was added, followed by 5 μg of an internal sterol standard. Upon addition of 4.5 ml 60% (w/v) KOH, the mixture was vortexed and heated at 75°C for 2 h in a water bath with gentle agitation to saponify steryl

esters. Samples were allowed to cool to room temperature and 4 ml petroleum ether was transferred to each sample using a glass pipette. Samples were then thoroughly vortexed for 15–20 sec followed by phase separation. The top phase was transferred to a 13x100 mm disposable glass culture tube and dried down under a stream of nitrogen, followed by resuspension in 250 μl heptane and transfer to an appropriate vial for GC analysis. To analyze the ability of the transformed yeast strains to grow under low oxygen condition, yeast colonies from a 2-day old plate were picked and resuspended in 1.5 ml YES media and the solution diluted to 0.16 $OD_{600}$. Next, a serial 1:5 dilution was prepared and 3 μl of cells of each dilution were spotted on YES plates containing 1.6 mM $CoCl_2$, and colony formation was analyzed after incubation at 30˚C for 5 days.

## Genomic insertion of TgCYP450mt-HA and conditional expression of TgCYP450mt

BLAST searches in the *Toxoplasma* database (www.toxoDB.org) identified a cytochrome p450 superfamily protein that was putatively mitochondrial (TgCYP450mt, TGME49_315770). For in-situ C-terminal tagging of the CYP450 gene, primers CYP-3HAgRNA.Mutag-R1 and CYP-3HAgRNA.Mutag-F1 (S1 Table) were used to introduce the protospacer tgaatcgccgttctgtgaca into pSAG1::Cas9-U6::sgUPRT (plasmid #54467 from Addgene; [89]) to generate pSAG1::Cas9-U6::sgCYP4503HA. Homology regions corresponding to the regions upstream of the protospacer (GCCTGCTGCCTGTGGTTCTTTCCTCTGAATCTTCTGCCAT) and the region downstream (TCCCGACAAGCCTGTCATGCTTCGTTTCAAACCTCGGGCG) were added to a base primer (primers HACATRT-R1 and HACATRT-F1) to amplify the 3xHA tag and chloramphenicol acetyl transferase (CAT) resistance cassette from pLIC-3xHA-CAT provided by S. Moreno (University of Georgia, Athens). One μg of the linearized PCR product and 5 μg of pSAG1::Cas9-U6::sgCYP4503HA were transfected into RH Δku80 TATi parasites. Transfectants were selected with 20 μM chloramphenicol immediately after transfection for population subcloning. Genomic DNA was extracted from isolated clones and screened by PCR using primers that encompass the transition from the endogenous CYP450 gene to the start of the inserted 3xHA tag and CAM resistance cassette (primer CYP-3HAV-F1) and downstream of the 3xHA tag and CAT resistance cassette (primer CYP-3HAV-R1), resulting in a 2.2-kb PCR product. The resulting PCR reaction verified that 7 out of 14 tested subclones were positive for integration of the 3xHA tag and CAT resistance cassette. Clones were further confirmed by Western blot and genomic sequencing.

For conditional knockdown of *CYP450*, primers CYP-gRNA.Mutag-R1 and CYP-gRNA. Mutag-F1 were used to introduce the protospacer caaggcagccaacacaccaa into pSAG1:: CAS9-U6::sgUPRT to generate pSAG1::Cas9-U6::sgCYP450. Homology regions corresponding to the region upstream of the protospacer (GTCACAGAGATTGAGAGGAACCCAC GTGCTTTGTTTGCAT) and the region at the beginning of the translational start codon (TCGCGTGTGAGACTGCGGTGACGTTGAAGTCGT) were added to a base primer (primers PRHRT-R1 and PRHRT-F1) to amplify the promotor insertion cassette pSAG4-tTA (provided by S. Moreno, UGA) with a tetracycline transactivator (tTA) upstream of a SAG4 promotor. This results in clones stably expressing a tetracycline-associated conditional transcription system (TATi) with replacement of the endogenous *CYP450* promotor via homologous recombination, causing *CYP450* expression regulated upon TATi binding to the tetracycline response element in the presence of anhydrous tetracycline. One μg of the linearized PCR product and 5 μg of pSAG1::CAS9-U6::sgCYP450 were transfected into RH Δ*ku80 TATi* parasites. Pyrimethamine (10 μM) was added to the parasite culture 24 h post-transfection for population subcloning. Genomic DNA extracted from isolated subclones were verified

by PCR using primers that are complimentary to the integrated inducible promotor (primer CYP-PIV-R1) and the original 5' UTR of the *Cyp450* gene (primer CYP-PIV-F1) with proper integration resulting in a 4.1-kb PCR-product. Clones were further confirmed via sequencing and inducibility confirmed by Western blot. Anhydrotetracycline at 1 μg/ml was added to knock-down the expression of CYP450.

## Phenotypic characterization of ΔTgMAPR and iΔTgCYP450mt

Plaque assays were performed as previously described [85]. Briefly, confluent HFF monolayers in a 6 well-plate were infected with 150 parasites each, cultured for 7 days without perturbation, fixed with -80°C methanol and stained with crystal violet for 10 min followed by 2 washes with PBS. Plates were scanned and mean plaque area measured using Volocity software. To evaluate the parasite invasion efficiency, confluent HFF grown on 24-well cover slips were infected with $10^7$ parasites for 20 min. Coverslips were washed 3-times to remove unattached/uninvaded parasites, fixed with 4% formaldehyde, blocked with 3% BSA in PBS, and external (attached) parasites stained with mouse-anti-SAG1 antibody (secondary anti-mouse Alexa-594). Next, cells were permeabilized with detergent, blocked, and subjected to immunofluorescence to stain internal (invaded) parasites using rabbit-anti-SAG1 antibody (secondary anti-rabbit Alexa-488). Invasion efficiency was calculated by determining the ratio of red versus green parasites [47] in 50 randomly selected fields observed at 40x magnification using a Nikon Eclipse 90i microscope (Nikon, Melville, NY) equipped with an oil-immersion plan Apo 100x NA 1.4 objective and a Hamamatsu GRCA-ER camera (Hamamatsu Photonics, Hamamatsu, Japan). Intracellular parasite replication was assessed by [$^3$H]uracil incorporation assay [85]. HFF cells in a 24-well plate were infected with 50,000 parasites each for 24 h prior to exposure to 1 μCi of [$^3$H]uracil for 2 h. Cells were washed twice with ice-cold PBS, incubated in 10% TCA on ice for 15 min, followed by vigorous shaking in 0.2N NaOH for 15 min. Samples were transferred into vials containing scintillation cocktail and radioactivity measured using a scintillation counter (Multipurpose Scintillation Counter, Beckman, Brea, CA). Egress assays were performed as described [85]. Briefly, confluent HFF cells in a 6-well plate were infected for 24 h, and parasite egress after exposure to A23187 (1 μM) was measured by time-lapse microscopy (Nikon Eclipse TE200 microscope). Images were acquired with a Spot RT CCD camera (Diagnostic Instruments, Sterling Heights, MI) and SPOT advanced software.

## Cell cycle analysis

VERO cells were infected with ΔTgMAPR, ΔTgMAPR$^{ad}$, parental or complemented strains at an M.O.I of 10 and washed 2 h p.i. to synchronize the infection. After 15 h, infected cells were exposed to the calcium ionophore A23187 at 4 μM for 10 min to induce parasites egress. Parasites in the supernatant were collected, washed by centrifugation, filter-purified, fixed in ice-cold ethanol (70%) with shaking during 5 min, pelleted at 3000×*g* and resuspended in PBS at a final concentration of $10^7$ parasites/ml. Parasites were then incubated with propidium iodide (PI) solution made of 0.01 mg/ml PI, 0.01 mg/ml DNAse-free RNAse A (Ambion Corp., Austin, TX), 0.1% Triton X-100, 1 mg/ml sodium citrate (pH 7.5) at 4°C overnight. Flow cytometry analysis was performed on a FACScan flow cytometer (Becton Dickinson, Mountain View, CA). Cell cycle distribution pattern was assessed using FlowJo software (Tree Star, Inc.); G1 and S peaks were defined using the Dean-Jet-Fox model. All assays were performed in triplicate and percentages of parasites in each phase of the cell cycle were compared for statistical significance between control and test groups using a two-tailed t test.

## Transcriptome sequencing

ΔTgMAPR and ΔTgMAPR[ad] parasites were grown in fibroblasts for 24 h, syringe-released by repeated passage through a 20 G1 needle then by a 22 G1 ½ needle, collected by centrifugation and resuspended in PBS buffer. Total RNA for the two strains from 3 independent parasite preparations was extracted using a Qiagen RNeasy kit. The total RNA was shipped to Novogene Corporation (Sacramento, CA) to be processed and analyzed. First, total RNA was converted to sequencing read libraries using the NEB Next Ultra kit (Illumina). The libraries were subjected to paired-end sequencing using the Novaseq 6000 with a read length of 150 bp. Each sample was sequenced to a depth of at least 20 million reads. The sequencing reads per sample were trimmed and mapped to the genome of *Toxoplasma* GT1 strain (release 39) for gene differential expression.

## LC-MS targeted lipidomics analyses for steroid detection

Lipidomics of *Toxoplasma* were carried out on HFF grown to confluency for 7 days in forty T-75 flaks. After confluency, each flask was washed 3x in PBS and the media replaced with DMEM-HG containing 1% delipidated serum (Calico Biolabs, Inc., Pleasanton, CA). Each flask was then infected with $2x10E^7$ tachyzoites and allowed to grow for 3 days or until the monolayer was fully lysed, whereby the media containing host cell debris and extracellular tachyzoites were collected and filtered 2-times sequentially through a 10 μm, 8 μm, 5 μm, then 3μm tack etched nucleopore Whatman filter. Remaining filtered supernatant was centrifuged at 1,800 rpm for 20 min to pellet parasites, with the resulting supernatant collected and used to wash each of the Whatman filters used for parasite purification, creating the host cell debris sample used as a control to quantify the potential contribution of metabolites specifically released from the lysis of host cells. The purified parasite pellet was then resuspended and washed 5-times in PBS. The resulting parasite sample was centrifuged for a final time and the remaining PBS aspirated before the final resuspension in 1 ml cold PBS and transfer to a 2 ml microcentrifuge tube. This suspension was centrifuged at 4500x*g*, media aspirated, and resulting pellet weighed before snap freezing in liquid nitrogen and storage in -80˚C overnight. These preparations of purified extracellular WT *Toxoplasma* and culture supernatant debris were sent to Creative Proteomics (Shirley, NY) for quantitative measurement of steroids using Agilent 1260 LC-MS 6420A, Poroshell 120 EC-C18 column (Column temperature: 35˚C; Mobile phase: A: 0.1% formic acid water; B: methanol; Flow rate: 0.3 ml/min) and the following standards: Aldosterone, Cortisone, Corticosterone, 11-Deoxycortisol, 19-Norandrostenedione, β-Nortestosterone, 4-Androstene-3,17-dione, 11-Deoxycorticosterone, 2-Methoxyestradiol, 17-Hydroxypregnenolone, Dehydroepiandrosterone, 17α-Hydroxyprogesterone, Androstanediol, trans-Androsterone, Dihydrotestosterone, cis-Androsterone, Pregnenolone, Hydrocortisone, Estradiol, Estrone, Testosterone, Estriol and Progesterone.

## LC-MS metabolomics analyses for steroid content analysis

To assess the steroid content in ΔTgMAPR and iΔTgCYP450mt relative to WT parasites, extracellular tachyzoites were purified as described above after 3 days of cultivation in HFF, with 2 days of ATc treatment for iΔTgCYP450mt parasites. After storage in -80˚C overnight, metabolites were methanol extracted from the sample by resuspending the frozen pellet in 1 ml 80% methanol, vortexing and sonicating on ice at 40% amplitude for 1 sec on 2 sec off-intervals for 5 min. The sonicated samples were centrifuged again at 4500x*g*, whereby the pellet was preserved for protein quantification via Nanodrop absorbance at 280 nm and the metabolite containing supernatant was added to a speed vac and spun for 120 min until the volume of the supernatant reached a final volume of 200 μl. The 200 μl metabolite suspension was snap

frozen in liquid nitrogen and then lyophilized overnight. Lyophilized metabolites were resuspended in 50% acetonitrile in water, vortexed, then gently rocked for 2 h at 4˚C. Samples were spun down at 14,000 rpm at 4˚C, with 100 μl of resulting supernatant transferred to a plastic vial for LC-MS. Quality control samples were made by mixing 4 μl of 500 μg/ml [$^{13}C_5$]glutamine in 50% acetonitrile. Intact HFF and the media with 1% delipidated serum used to cultivate parasites for lipidomic analysis were also subjected to metabolite extraction with resulting metabolites used for comparative analysis. Measurements were taken on a Thermoscientific Q Exactive Plus Orbitrap Mass Spectrometer using the following standards: Aldosterone, Cortisol, Cortisone, Corticosterone, DHEA and Progesterone. Normalization of raw intensity values were calculated by subtraction of background intensities measured for each metabolite and the creation of a scaling factor value calculated by dividing each sample's protein concentration by the lowest protein concentration detected. Raw intensity values were multiplied against each sample's scaling factor to calculate normalized raw intensities.

### Fluorescence microscopy

Light and epifluorescence microscopy were performed on infected cells seeded on sterile coverslips in 24-well culture dishes. Immunofluorescence assays (IFA) on infected mammalian cells were performed as described previously [8], with a 5 min permeabilization step with 0.3% TritonX-100 in PBS following fixation with 4% formaldehyde plus 0.02% glutaraldehyde in PBS for 15 min. Coverslips were mounted using ProLong Diamond Antifade Mountant to minimize bleaching during microscopy. Slides were viewed using an oil immersion plan Apo 100x (NA1.4) objective, a Zeiss AxioImager M2 and a Hamamatsu Orca-R2 digital camera. Mander's Overlap Coefficients (MOC) were calculated using Volocity software (Quorum Technologies) along with a product of the difference of the means image map (PDM), which is generated by identifying voxels where each signal is above its mean value after thresholding for background. Thresholds were set by measuring, using an ROI, the fluorescence intensity of each channel in a region of the nucleus without staining for either signal (background). These background values (per image) were used to calculate MOC and PDM values of the corresponding individual infected cells. The positive PDM shows voxels where both fluorescence signals are above the mean and show positive correlation. Proximity ligation assays (PLA) were carried out using Proximity Ligation assay kit components for Far-red fluorescence detection using anti-mouse plus and anti-rabbit minus probes. The Duolink PLA fluorescence protocol was followed according to the manufacturer instructions (Sigma), using the mouse anti-TgMAPR antibody at 1:250 dilution, mouse or rabbit anti-HA antibody at 1:1000 dilution and rabbit anti-HSP70 antibody at 1:500 dilution.

### Electron microscopy

For ultrastructural observations of *T. gondii*-infected cells by thin-section transmission electron microscopy (EM), infected cells were fixed in 2.5% glutaraldehyde (Electron Microscopy Sciences, Hatfield, PA) in 0.1 mM sodium cacodylate and processed as described [8]. Ultrathin sections of infected cells were stained before examination with a 7600 EM under 80 kV, equipped with a dual AMT CCD camera system. For immunoEM gold staining of TgMAPR-HA, infected fibroblasts were fixed in 4% paraformaldehyde (Electron Microscopy Sciences) in 0.25 M HEPES (pH 7.4) for 1 h at room temperature, then in 8% paraformaldehyde in the same buffer overnight at 4˚C. Cells were then infiltrated, frozen and sectioned, immunolabeled with mouse anti-HA antibody (1:100 in PBS/1% fish skin gelatin), then with protein A coupled to 10 nm gold particles and processed as previously described [5].

## Supporting information

**S1 Fig. Analytical results of steroid detected in *Toxoplasma*: Regression Equations and Calculation Results (from Creative Proteomics).**
(PDF)

**S2 Fig. Protein sequence alignment of CYP450 homologs.** (A) Sequence comparison between TgCYP450mt with closest human homologs that share the motif for heme-binding (red box). In blue are the six CYP450 localized to mitochondria. (B) C-terminal regions of CYP450 homologs in Sarcocystidae as shown for *Sarcocystis neurona* (SN3_00900360), *Cystoisospora suis* (CSUI_005351), *Besnoitia besnoiti* (BESB_078830), *Neospora canimum* (NCLIV_058440), *Toxoplasma gondii* (TGME49_315770) and *Hammondia hammondi* (HHA_315770). In green: helix K domain to stabilize the protein core and in yellow: heme-binding loop to position the iron atom in the heme.
(PDF)

**S3 Fig. Generation of parasite strains.** See detailed description of molecular constructs in the Materials and Methods section. (A) Strategy for in-situ C-terminal tagging with 3xHA with the red region denoting the stop codon and the dotted black line the CRISPR/Cas9- mediated cut, verified by PCR and Western blotting using an anti-HA antibody and identifying 7 positive clones (shown in the lanes). (B) Promotor displacement strategy for conditional expression of the *TgCYP450mt* gene under the control of tetracycline-dependent SAG4 promotor, confirmed by PCR and Western blotting using anti-HA antibody showing down-expression of TgCYP450mt-HA after exposure to 1μg/ml of anhydrous tetracycline for 24 h (~70% decrease) and 48 h (~90% decrease), compared to no added tetracycline. Loading control anti-TgTubulin antibody was used to assess protein expression levels and validate Western blot analysis. (C) Strategy for direct gene deletion of TgMAPR via double homologous recombination as confirmed by PCR and Western blotting using anti-TgMAPR antibody. (D) Strategy for introduction of the TgMAPR gene on ΔTgMAPR parasites. Successful gene complementation was probed by PCR, Western blotting using anti-HA antibody and IFA using two different anti-HA antibodies.
(PDF)

**S4 Fig. Protein sequence alignment of MAPR homologs.** (A) In Sarcocystidae, shown for *Neospora caninum* (NCLIV_060960), *Besnoitia besnoiti* (BESB_027250), *Toxoplasma gondii* (TGME49_276990; TgMAPR) and *Hammondia hammondi* (HHA_276990). Shown are N-terminal sequences. Highlighted in green: one predicted transmembrane domain and highlighted in turquoise: the MAPR/Cytb5 domain with the sole axial tyrosinate residue (blue box) for the iron chelation of heme. (B) Sequence comparison between TgMAPR, human PGRMC1 (HsPGRMC1; Swiss-Prot O00264) and *Schizosaccharomyces pombe* (SpDap1; SPAC25B8.01) that share the MAPR/Cytb5 domain (highlighted in turquoise); the conserved tyrosine and aspartic acid residues required for heme binding (blue boxes); potential phosphorylation sites (green boxes); potential SH2 domain (green boxes at the C-terminus) and SH3 domain (fuchsia box). The three residues in yellow typify the PGRMC/Dap1 proteins, with two present in TgMAPR.
(PDF)

**S5 Fig. Mitochondrial localization of TgMAPR.** (A) Immunofluorescence for localization of TgMAPR in mitochondria in HeLa cells transfected with TgMAPR-HA; anti-HA (green), Mito-T (red). (B) Double IFA on TgMAPR-HA-expressing *Toxoplasma* using anti-HA and anti-TgMAPR antibodies showing colocalization with positive PPC. (C) Western blots on

freeze-thaw lysates from TgMAPR-HA-expressing Toxoplasma using anti-TgHA antibody revealing 2 bands corresponding to monomeric and dimeric forms of TgMAPR-HA. (D) ImmunoEM on TgMAPR-HA-expressing *Toxoplasma* using anti-HA antibody confirming gold particles on mitochondrial membranes (white arrow on the outer membrane and black arrow on the inner membrane). (E-F) Double IFA on TgMAPR-HA-expressing *Toxoplasma* using anti-HA and anti-HSP70 antibodies showing colocalization. Values of PCC and MOC are shown. Arrows in F show area enriched in TgMAPR.
(PDF)

**S6 Fig. Functional complementation of SpDap1Δ with TgMAPR.** (A) Expression of TgMAPR in yeast *SpDap1Δ* under hypoxic conditions. Panel i: Western blots of TgMAPR-*myc*-expressing *Dap1Δ* of *S. pombe* using anti-myc antibody, verifying TgMAPR expression, and 3xHA-SpDAP1 expressed in *Dap1Δ* of *S. pombe* using anti-HA antibody, verifying the complementation. Panel ii: Yeast growth assay at 30˚C for 3 days in medium containing $CoCl_2$, showing rescue of growth of dap1Δ upon expression of either SpDap1 or TgMAPR, with comparable growth to the yeast parental strain (panel b). (B) Sterol profiles in yeast *SpDap1Δ* expressing TgMAPR. Ergosterol biosynthetic pathway in yeast showing the role for Dap1p in activating Erg11p is shown in the inset. Sterols extracted from log-phase WT, parental, *Spdap1Δ* yeast expressing Dap1 (positive control, vector alone (negative control) or TgMAPR were analyzed by gas chromatography and GC-MS. Data are the average of three independent replicates ± SD.
(PDF)

**S7 Fig. Gating strategy to select parasite populations based on side-scatter pulse height on the FL2 total area channel.** Purified parasites were labeled with propidium iodide to analyze cell cycle based on DNA content for experiments shown in Figs 4C(A) and 5E(B).
(PDF)

**S8 Fig. RNA-Seq comparison between ΔTgMAPR^ad^ and ΔTgMAPR^<7wks^ parasites.** Volcano plot revealing 11 genes having increased expression (red) and 7 with decreased expression (green) in ΔTgMAPR^ad^, with statistical significance less than 0.05, relative to ΔTgMAPR^<7wks^. The green and red dashed lines represent the borderline of Log2 fold change of 0.5 in gene transcripts, and the genes above the black dashed line had padj values of statistical significance below 0.05. Each sample was sequenced in duplicate for statistical comparison. Shown is the list of the 18 genes that are differentially expressed, with some identified in SWISS-PROT and ToxoDB.
(PDF)

**S9 Fig. Absence of interaction between HSP70 and TgCYP450mt or TgMAPR in the mitochondrion.** Top. *In situ* interaction of TgCYP450mt and TgMAPR showing red fluorescent signal in PLA (see Fig 8B). Fluorescence microscopy of TgMAPR-HA- or TgCYP450-HA-expressing *Toxoplasma* using a PLA showing no fluorescent signal using primary antibody alone (anti-HA for each strain or anti-HSP70) or both primary antibodies (anti-HA for each strain and anti-HSP70). Nuclei staining by DAPI.
(PDF)

**S10 Fig. Schematic representation of steroid biosynthesis from cholesterol in Vertebrates.** In blue: mitochondrial enzymes with functions compatible to TgCYP450mt.
(PDF)

**S1 Table. Cloning primers used for the study of TgMAPR and TgCYP450mt.**
(PDF)

**S1 Data. Supplemental Data include Excel spreadsheets with the raw numerical data for graphs.**
(PDF)

## Acknowledgments

The authors gratefully thank the members of the Coppens laboratory for helpful discussions during the course of this work, and the technical staff from the Johns Hopkins Microscopy Facility (M. Delannoy, B. Smith) and the Yale Microscopy Facility (K. Zichichi). We also thank Peter Espenshade (Johns Hopkins University) for the gift if *S. pombe* dap1Δ mutant. This work relies on VEuPathDB.org and we thank all contributors to this resource.

## Author Contributions

**Conceptualization:** Beejan Asady, Vera Sampels, Isabelle Coppens.

**Data curation:** Beejan Asady, Vera Sampels, Julia D. Romano, Anne Le, Isabelle Coppens.

**Formal analysis:** Beejan Asady, Isabelle Coppens.

**Funding acquisition:** Isabelle Coppens.

**Investigation:** Beejan Asady, Vera Sampels, Pratik Khare, Isabelle Coppens.

**Methodology:** Beejan Asady, Vera Sampels, Julia D. Romano, Jelena Levitskaya, Bao Lige, Pratik Khare, Isabelle Coppens.

**Project administration:** Isabelle Coppens.

**Resources:** Anne Le, Isabelle Coppens.

**Supervision:** Beejan Asady, Anne Le, Isabelle Coppens.

**Validation:** Beejan Asady, Vera Sampels, Anne Le, Isabelle Coppens.

**Visualization:** Beejan Asady, Vera Sampels, Isabelle Coppens.

**Writing – original draft:** Isabelle Coppens.

**Writing – review & editing:** Beejan Asady, Vera Sampels, Julia D. Romano, Jelena Levitskaya, Bao Lige, Pratik Khare, Anne Le, Isabelle Coppens.

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
