## [Decision Letter · Decision Letter 0]

13 Feb 2023

Dear Dr. Coppens,

Thank you very much for submitting your manuscript "Steroids in Toxoplasma gondii: Synergistic activities between a steroidogenic CYP450 enzyme and a Membrane-Associated Progesterone Receptor (MAPR) homolog in the parasite mitochondrion" for consideration at PLOS Pathogens. As with all papers reviewed by the journal, your manuscript was reviewed by members of the editorial board and by several independent reviewers. In light of the reviews (below this email), we would like to invite the resubmission of a significantly-revised version that takes into account the reviewers' comments.

I am returning your manuscript with three reviews. The following reviewer concerns must be addressed prior to resubmission:

1) Reviewers were concerned about that execution and interpretation of experiments with AG-205. No direct evidence was provided to demonstrate that this compound was specifically targeting TgMAPR. The observed phenotypes could be due to inhibition of MAPR in the host cells or off target effects in the parasite.

2) The organization and clarity of figures and the frequent “typos” made the manuscript  difficult to navigate and detracted from the data itself. Consider reorganizing the figures to eliminate redundancy.

3) More thorough quantification and statistical analysis of knockout/knockdown phenotypes.

In re-writing, authors should also temper their conclusions and discuss the possibility that the proteins investigated in this study may have functions independent of steroidogenesis, as pointed out by reviewer 3 or complete the additional experiments suggested by the reviewer to address this concern.

In addition to the revisions requested by the reviewers, please use color-blind friendly pseudo-coloring in all figures.

We cannot make any decision about publication until we have seen the revised manuscript and your response to the reviewers' comments. Your revised manuscript is also likely to be sent to reviewers for further evaluation.

Sincerely,

Aoife T. Heaslip, Ph.D

Guest Editor

PLOS Pathogens

Margaret Phillips

Section Editor

PLOS Pathogens

Kasturi Haldar

Editor-in-Chief

PLOS Pathogens

orcid.org/0000-0001-5065-158X

Michael Malim

Editor-in-Chief

PLOS Pathogens

orcid.org/0000-0002-7699-2064

Reviewer's Responses to Questions

**Part I - Summary**

Reviewer #1: This study provides convincing evidence and novel insights in the role of TgMAPR and TgCYP450 in steroidogenesis. Although the large amount of complementary data in this study is robust, the paper would greatly benefit from inclusion of additional controls, increased consistency, and polishing of the figures and text.

This study is the first to address steroidogenesis in Toxoplasma, a completely novel and surprising aspect of Toxoplasma biology. Authors provide an in-dept description of the TgMAPR and TgCYP450 knockdown or knockout phenotypes and their potential roles in the steroid pathway in the parasite mitochondrion. The paper includes many complementary experiments and a tremendous amount of work that support the authors claims. While the experimental design is robust and the data is convincing, there are some shortcomings. The authors could have paid more attention to details and could have been more consistent in the text and figures. Because of these minor but frequent errors and inconsistencies, the general impression of the paper is somewhat chaotic. Additionally, some experiments could be improved by adding controls. However, if these issues are addressed, this manuscript can provide an excellent foundation for future studies on the role and biosynthesis of steroids in Toxoplasma and other pathogenic unicellular eukaryotes.

A major strength of this paper is the amount of data generated and the diversity of approaches to address the roles TgMAPR and TgCYP450 in steroidogenesis. Detailed experiments including light and electron microscopy, growth, mobility, invasion, and egress assays and more, convincingly describe TgMAPR and TgCYP450 phenotypes. The extensive in vitro work is nicely complemented by the in vivo virulence assays. The presence of steroids in Toxoplasma and a role of TgMAPR and TgCYP450 in the steroid production pathway is convincingly demonstrated by the LC-MS analysis.

The major point of critique for this paper is the general “untidy” or “hasty” impression that arises because of small, but frequent textual errors and discrepancies, accompanied by disorderly figures which often lack scalebars and proper alignment. The chaotic impression of the figures detracts to some degree from the data presented in them. Proper alignment of panel letters, images, graphs, and text with equal spacing and the inclusion of scalebars in all microscopy images could greatly improve the quality of the figures. Some examples are discussed in “minor points” section below.

For convenience of review, we would strongly encourage the authors to use (continuous) line numbering and place figures and tables in line with the text, and minimally the legends together with the figures. This prevents a lot of unnecessary scrolling and makes it easier to refer to specific sections in the text.

Reviewer #2: The manuscript by Asady and colleagues describes the identification of two mitochondrial proteins in the apicomplexan parasite Toxoplasma gondii with a role in steroid metabolism. The study provides some interesting insights into the importance of steroid metabolism in these parasites, which is not well understood. Although some key data are supported by the evidence the authors provide (the localisation of the two proteins to the mitochondrion, their importance for parasite proliferation and virulence, and their contribution to steroid metabolism), other data are not quantified, not properly controlled and/or provide limited insights (most notably, the consequences of the loss of proteins on various aspects of parasite cell biology, and the molecular target of the AG-205 inhibitor that they test). The authors should consider paring back their study to focus on conclusions that are well-supported by the data, and undertake a more thorough, quantitative, and properly controlled analysis of phenotypes.

Reviewer #3: Cholesterol is an essential component of cellular membranes, as well as a key precursor of signaling molecules including steroid hormones that coordinate growth, development and reproduction in mammals, and that are generated in a process known as steroidogenesis. Several microbes such as the human parasite Toxoplasma gondii lack the machinery to synthesize cholesterol, and thus must acquire it from the host cell. Yet, whether Toxoplasma possesses the machinery to process cholesterol into steroid signaling molecules, and whether such steroids impact parasite development remain unknown. In this study, the authors investigate steroidogenesis in Toxoplasma. They focus on two Toxoplasma genes, TgCYP450mt and TgMAPR, that have homology to membrane-bound cytochrome P450 oxidases and membrane-associated progesterone receptors, respectively. The authors present data that support that TgMAPR is a heme-binding protein that is localized to the parasite mitochondrion, required for growth, virulence in mice, and maintaining levels of several steroids including progesterone, corticosterone, DHEA, and cortisol. A characterization of TgCYP450mt revealed that it, like TgMAPR, localized to the mitochondrion where it interacted with TgMAPR. Parasites deficient for TgCYP450mt had even lower levels of steroids such as progesterone, corticosterone, DHEA, and cortisol, and were deficient for growth in vitro and virulence in vivo. The authors propose a model in which heme-mediated TgMAPR activation promotes TgCYP450mt activation, and the induction of steroidogenesis. Although the authors put forth the exciting possibility that Toxoplasma uses steroids as signaling molecules that promote replication and development—despite that it is a cholesterol auxotroph—several of the conclusions between the proteins they have studies, their role in steroidogenesis, and the role of steroidogenesis for Toxoplasma are correlative.

**Part II – Major Issues: Key Experiments Required for Acceptance**

Reviewer #1: Too little details are provided on how mouse infections were monitored and what humane endpoints were used to terminate the experiments, please elaborate.

Figure 1. The claim that TgMAPR forms a dimer, is not sufficiently supported. While TgMAPR-HA shows a small band corresponding to the expected mass of the dimer in Fig1A, this is all performed in an artificial overexpression context and the dimer band is not found when using the TgMAPR antibody. By using the available TgMAPR antibody and the freeze-thaw protocol used in the left panel, the authors might be able to show the presence of a dimer in the wildtype parasite line, circumventing possible artefacts arising from protein over-expression.

Figure 3 includes data showing phenotypes caused by the AG-205 compound, which is a ligand of mammalian PGMRC1. Although PGMRC1 shows homology with TgMAPR and therefore it would be logical to assume that AG-205 also targets TgMAPR, there is no evidence in the paper that supports this assumption. Moreover, overexpression of TgMAPR-HA did not alter the parasite sensitivity to AG-205. Therefore, it could be possible that the observed phenotype in figure 3 is due to the AG-205 targeting the host cell and therefore indirectly altering parasite growth and mitochondrial phenotype or the drug exerting its function through a different target in the parasite. Authors could provide evidence for TgMAPR being the AG-205 drug target by including the adapted TgMAPR KO in their assays. If TgMAPR is indeed the drug target, AG-205 should not influence the growth of the adapted TgMAPR KO parasites. If the growth is affected, it is likely that AG-205 works through a different mechanism and analysis should be changed accordingly.

Reviewer #2: 1. Figure 2. “Gold particles were detected both on the outer and inner membranes of the mitochondrion, including cristae” … “TgMAPR localizes to the outer and inner membranes of the parasite mitochondrion”. A protein that localizes to both the outer and inner mitochondrial membranes is very unusual. Without controls for the distribution of “representative” outer and inner membrane proteins, these data are not particularly convincing. The conclusion that the protein is a membrane bound one is reasonable (although would benefit from a more direct analysis – e.g. sodium carbonate extractions), but in the absence of proper controls or a more thorough analysis, the authors should temper their conclusions about the sub-mitochondrial localization of TgMAPR.

2. Figure 3. The AG-205 experiments are not well-justified or particularly convincing. The authors present no evidence that AG-205 is an on-target inhibitor of TgMAPR or any other protein associated with steroid metabolism. Without evidence that AG-205 is an on-target inhibitor of TgMAPR (or other protein involved in steroid metabolism), these data do not add much to the story. Additionally, the effects of AG-205 on cellular and mitochondrial morphology is interesting, but is not quantified in any way. The authors could consider undertaking immunofluorescence assays to examine mitochondrial morphology upon AG-205 treatment, or undertaking a more thorough quantification of the electron microscopy data (ideally with serial sections or an equivalent approach to demonstrate the overall shape of the mitochondria). They might also consider testing whether AG-205 treatment leads to changes in steroid abundances (as they do for the ∆MAPR and P450 knockdown strains in Figure 11).

3. Figure 3C-D. The statistical comparisons here are for each treatment vs the “no drug, 10% serum” condition. A more appropriate comparison, that controls for the effects of serum, would be between the drug-treated data and the no drug data for the “plus serum” samples (as they currently do), and for the drug-treated data and the no drug data for the “minus serum” samples.

4. Figure 5A. As with Figure 3, the proposed defects on MAPR knockout on endodyogeny and mitochondria requires quantification, especially given that there are some normal looking mitochondria in the mutant images as well (top cell, Figure 5D, panel d). Undertaking immunofluorescence assays to examine daughter budding and the overall morphology of the mitochondria (rather than just the ultrastructural details revealed by TEM) might give a clearer picture of how endodyogeny and mitochondria are affected by MAPR knockout.

5. Figure 5B. The DBA experiments require quantification. What proportion of vacuoles in the ∆MAPR strain are DBA positive?

6. Figure 7. As with Figure 5A, these data on morphological abnormalities in the “culture adapted” strain require quantification. e.g. what percentage of parasites have “upside-down” daughter buds? Are there gross morphological defects in rhoptries that match the vesiculation seen at an ultrastructural level? As with Figure 5B, the DBA experiment requires quantification.

7. Figure 9. The PLA assays would benefit from controls testing whether other mitochondrial proteins not expected to complex with TgCYP450mt or TgMAPR demonstrate an interaction. Alternatively, the authors could consider co-immunoprecipitation experiments testing whether TgCYP450mt and TgMAPR interact in parasites.

Reviewer #3: Major Point 1:

Role of steroidogenesis (SG) in Toxoplasma gondii:

Using TgMAPR KO and iKDs of TgCYP450mt that are deficient for growth and have decreased levels of several of the steroids measured, the authors argue that steroidogenesis is essential for Toxoplasma growth and proliferation. However, steroidogenesis is not required for the survival of a eukaryotic cell (most cells in the human body are for example non-steroidogenic); and it is possible that these proteins have functions independently of SG that contributes to the negative effects of their deletion.

1) TgMAPR is a receptor that in mammals is not essential for steroidogenesis — it is unclear why the authors investigate this protein and not STAR that is required to transport cholesterol into the mitochodnria that is required for SG?

2) Have the authors supplemented cells infected with SG-mutants (TgMAPR KO and iKDs of TgCYP450mt) with the steroids for which they are deficient? Does this rescue growth and or development?

3) Have the authors expressed heme-binding mutants and or enzymatic mutants of TgMAPR KO and iKDs of TgCYP450mt, respectively? How does this affect growth?

4) The authors report an ‘adapted TgMAPR KO. Does this parasite line have higher levels of the steroids deficient in the TgMAPR KO? This would be the expected result if steroids are essential for parasite proliferation.

5) TgMAPRKO parasites appear to form cyst-like structures that are positive for TRITC-lectin (fig. 5B). The authors conclude that these are not cysts because they are BAG1-deficient but have other markers of cyst formation been examined? Has cyst formation in surviving mice been examined? Have the authors examined cyst-formation in Type II MAPR KOs?

Major Point 2: Effect of TgMAPR and TgCYP450mt on steroidogenesis:

The authors show that parasites deficient for TgMAPR and TgCYP450mt have decreased levels of various steroids (Fig. 11). How was this experiment conducted? Given the dramatically different growth rates, how can the authors be sure this is reflecting a difference in steroid synthesis rather than development-stage synthesis (i.e. more SG occurs at later stages in the lytic cycle). Furthermore, how can the authors be sure that their deficiency is directly affecting SG rather than indirectly affecting mitochondria? Can enzymes more downstream in SG be ablated? Can aromatase inhibitors be tested for their effects on Toxoplasma SG and growth?

**Part III – Minor Issues: Editorial and Data Presentation Modifications**

Reviewer #1: Title:

• The title is hard to digest, especially if the reader is yet unaware of the context of the paper. Authors may consider making it more concise. It is also not clear what “synergistic” refers to in this context since no clear synergy is demonstrated in this paper or referred to in the text.

Abstract and author summary

• “We engineered a knockout strain for conditional expression of CYP450 and iΔTgCYP450mt parasites have severe growth defects in cultured cells.” We recommend changing to “conditional knockdown”.

Introduction

• The introduction is fairly long and could be written more concise. Authors should re-evaluate what information is necessary to comprehend the paper. E.g. most information in the second paragraph of the introduction is not necessary for the reader to understand the rest of the paper.

• From the introduction it is not very clear what the role of steroids could be in T. gondii and therefore why authors would be interested in looking at steroidogenesis to begin with.

• Authors use gene IDs from both ME49 and GT1 (TGME49_315770 and TGGT1_276990) and it is unclear why and if the blast search was also done with these two background strains.

Results

• All figures:

o Panel letters should be aligned.

o Images, graphs and text should be properly aligned with equal spacing.

o All microscopy images should include a scale bar.

o Different images within a panel should not have similar labeling as the panels themselves (ABC, abc), but should use two different labeling systems (e.g. ABC and i, ii, iii).

o Labelling of channels and structures within micrographs should be consistent.

o With light microscopy images, the intensity of the separate channels should be the same as the intensity in the merge channel (in e.g. Fig2A and Fig8B Mitotracker signal).

o Bar graphs should have a consistent layout throughout the paper.

• P7: “ancestral MAPR gene” There is no evidence for TgMAPR being ancestral only that it branched early, but it has likely evolved to similar levels if not more extensively than other MAPR genes.

• P7: “Displaying various subcellular localizations, MAPR proteins fulfill a broad range of functions related to cell survival, including cholesterol synthesis and homeostasis, steroid hormone production and signaling, and resistance to DNA damage-induced stress (Ghosh et al., 2005).” This information could be moved to the introduction.

• Fig1. Panel C. Authors could explain the in-gel heme assay shortly in the text. As a reader not familiar with the assay you would have to go to the reference paper from 1993 to understand.

• Fig1. Panel C. Should not be on top of the graph in panel B.

• Fig1. Panel D. The image of the column is misleading. At first glance, the picture of the column seems a smear on the SDS-Page gel, as it is underlined with His-TgMAPR and seems in line with the anti-His gel images. Picture should be separated from the gel images and enlarged, so it is clear the reader is looking at a picture of a column.

• Fig1. Panel D. In our experience the Qiagen Ni2+-NTA column also sometimes turns brown when (over)loaded with non-heme binding proteins. Making this questionable as a line of evidence, particularly without any controls leading to differently colorings.

• Fig2. Panel B. The utility/purpose of overexpression of TgMAPR-HA in HeLa cells is not clear to us and could be omitted.

• Fig2. Panel C. The images are very small, a zoom-in could be used.

• Fig2. Panel D. IMM/OMM localization is done with overexpressed HA-tagged MAPR, so dual localization could also be a consequence of overexpression. Indeed, there are differences in localization of HA and anti-MAPR signals, which the authors fail to address in the text.

• Fig3. Panel A. Similar LUT should be used in all images (all gray or all color).

• Fig3. Panel B. Labels are not explained. For comparison host cell mitochondrion in untreated should also be shown.

• P11: KO mutant should be FigS3. In this Figure, even though the Ab stain convincingly demonstrates absence of MAPR, for completeness PCRs demonstrating absence of WT contaminants should be added.

• Fig5. Panel C. Please explain what the different colors in the graphs represent.

• Fig6. Panel F. Unusual display of mouse survival/death. More accurate would be to display in steps (as in Fig10) not diagonal lines as this is a binary read out.

• Fig7. Panel A. image c. The square is highly unusual and looks like an imaging artifact. Please explain.

• Fig8. The additional localization with MAPR Ab as opposed to HA antibodies can also be observed in Fig2 but was not commented on. This might be Ab specific effects, overexpression appears less likely the culprit. In not all cases does the “subdomain” coincide with Mitotracker, suggesting possible dual non-mito localization (possibly apicoplast?)

• Fig9. Panel A. Please provide the DAPI channel for the right-side panel. Furthermore, the background of the left-most panel seems to be different compared to the middle DAPI stains. It could be that different LUT settings are used between the different panels, please make sure that all LUTs are the same for all panels.

• Fig9. Panel B. It would be good to include another mitochondrial membrane protein as negative control to show reduced levels of interaction (if indeed, otherwise the assay would simply hint at close proximity in the membrane but not necessarily functional interactions) in which case other approaches (Co-IP, blue-native PAGE, etc) may be more suitable.

• P16. “possibly forming stable conjugates in the parasite mitochondrion.” We feel that there is not enough evidence for this statement.

• Fig. 10B. The accompanying thext indicates that all animals survive in the controls, whereas one animal actually dies for the +Atc control. This is merely a textual error which does not influence the conclusions.

• Fig11. Although this figure convincingly shows a decrease of different steroids in the TgMAPR KO and TgCYP450 iKD line, claims could be strengthened even further by including the complemented line and/or the WT treated with AG-205.

Discussion

• Fig12. There is not enough evidence for the model (see comments Figure 1).

• P19-20. There is no need to repeat so much of the discussion of the paper exploring CYP450 evolution as none of the work here other than identifying the gene contributes to this discussion much.

Materials and methods

• Primer sequences should not be mentioned in the text but can all be included in a supplementary table.

Reviewer #2: 1. Figure S1A. The alignment of the Neospora MAPR homologue to the other MAPR homologues from Sarcocystidae appears to diverge from approximately residue 184 of the Neospora protein, which corresponds to the splice site between the third and fourth exon. A tBLASTn search of the T. gondii MAPR homologue against the Neospora genome reveals the presence of a highly conserved region on the Neospora genome that corresponds to the 4th exon/C-terminus of the T. gondii MAPR protein. I suspect that the existing gene model for Neospora MAPR is probably incorrect. Consider revising the Neospora sequence or removing this from the alignment.

2. Figure S1B. Given the authors are testing for complementation of a S. pombe DAP1 knockout in Figure S2A, they could consider including the S. pombe homolog in this alignment.

3. Figure S3A. The position of some of the primers used in the PCR screening is a little unclear. Specifically, where do the ProgR-5UTR-F1 and ProgR-3UTR-R1 primers bind in the diagram? What are the “gene-specific” primers and where do these bind? Ideally, the authors would have performed a PCR testing for retention of the native TgMAPR gene in the parental strain (since an alternative explanation for the lack of bands on the parental lanes is that no DNA was present).

4. Figure 1A. “Membrane association of TgMAPR was verified by Western blots using anti-TgMAPR antibody.” These samples were solubilized in Triton X-100, which will solubilize membranes. The supernatant fraction will therefore contain both membrane-bound and soluble proteins, so this experiment isn’t testing whether the protein is membrane associated. If the authors mean that the protein being in the pellet fraction following freeze-thawing is evidence of its membrane association, then they need to include controls using antibodies against proteins that are membrane associated and not membrane associated to validate that this fractionation approach is valid.

5. Figure 1A legend. “Freeze-thaw (right) and detergent-solubilized (left) lysates” – the authors have left and right mixed up here.

6. Figure 1A. “Selective detergent cell separation of wild-type (WT) parasites and TgMAPR-HA-expressing Toxoplasma with TritonX-100 yielded a 28-kDa protein band in the solubilized fraction (Fig. 1B).” I think the authors mean to refer to the right panel of Figure 1A.

7. Figure 1C. If the TgMAPR experiment is on purified protein, there seems to be a lot of high molecular mass label. Can the authors account for this?

8. Figure 1D legend. “purified 6xHis-TgPGRMC1” – the authors have referred to this as TgMAPR elsewhere in the manuscript.

9. Figure S3D. What are the different lanes in the PCR analysis? Different clones that were screened? This should be described in the figure legend.

10. “TgMAPR-HA expressed under the control of the tubulin promotor localized to the parasite mitochondrion as expected (Fig. S3B)” Is this IFA with two anti-HA antibodies? Although the localisation clearly looks to be mitochondrial, the authors would ideally use a known mitochondrial marker for co-labelling to conclude this.

11. Figure 2E. "… gold particles decorated the inner membrane with 25% of the gold labeling density present on cristae (Fig. 2F).” The authors mean to refer to Figure 2E here.

12. Page 10. “mammalian PGMRC1” – I think the authors mean PGRMC1 (also a couple of other instances of this elsewhere in the manuscript)

13. “While some replication profiles with nascent daughters were observed with 4 individual parasites per PV (Fig. 5D, panel c)” – here and elsewhere in this paragraph, the authors mean to refer to Figure 5A.

14. Figure 5C and 6E. For clarity, consider defining the green and blue colors on the histogram in the figure legend.

15. Bottom paragraph, page 15. This paragraph is lacking relevant references (e.g. evidence for the functions of the various CYP450 proteins mentioned).

16. Figure 11. These are interesting data. Were the authors able to detect cholesterol as well? If so, these data might be worth including, since you might expect normal (or perhaps even increased) levels of cholesterol in parasite strains defective in downstream steroid metabolism.

17. Discussion, page 21. “Toxoplasma … also has a plant-like heme biosynthesis pathway”. Not strictly true. Although some enzymes do indeed function in the plastid (like in plants), the initial reaction in the pathway is very different from that in plants, and the later steps in the pathway occur in the cytosol and mitochondrion as in eukaryotes lacking plastids.

18. Methods, p. 27. “the endogenous PGRMC1 gene”. Do the authors mean to refer to the TgMAPR gene here?

19. Methods, p. 31. For the antibodies used in the immuno-EM, the authors describe using a mouse anti-HA primary antibody with an anti-rat secondary antibody. Please clarify. Also, were the protein A gold beads pre-attached to the secondary antibodies used? What are the sources (commercial?) of these antibodies?

Reviewer #3: Minor:

The authors should use a protein control for the assay used to demonstrate heme-binding by TgMAPR or a TgMAPR mutant for the heme-binding domain.

Indicating the name of the protein being visualized would contribute to figure clarity in IFAs (i.e. rather than simply HA).

Although interesting, without an understanding of the mechanism of action of AG-205 in Toxoplasma, the AG-205 experiments contribute little to support SG in Toxoplasma and should be removed for clarity.

The characterization of the MAPR-deficient parasites and CYP450mt-KD parasites should be combined for clarity. As is, the authors characterize one gene that appears essential for SG, than do a very similar set of analyses for CYP450mt, which reads redundantly.

The authors could include a figure comparing enzymes present in mammals that are required for SG relative to those they have identified in the Tg genome / characterized in this study.

PLOS authors have the option to publish the peer review history of their article (what does this mean?). If published, this will include your full peer review and any attached files.

Reviewer #1: **Yes: **The Kooij-Pond

Reviewer #2: No

Reviewer #3: **Yes: **Lena Pernas
---

## [Decision Letter · Decision Letter 1]

30 Jun 2023

Dear Dr. Coppens, 

Thank you very much for submitting your manuscript "Function and regulation of a steroidogenic CYP450 enzyme in the mitochondrion of Toxoplasma gondii" for consideration at PLOS Pathogens. As with all papers reviewed by the journal, your manuscript was reviewed by members of the editorial board and by several independent reviewers. The reviewers appreciated the attention to an important topic. Based on the reviews, we are likely to accept this manuscript for publication, providing that you modify the manuscript according to the review recommendations.

There is disagreement among the reviewers in regards to the interpretation of the data. Reviewer 3 is not satisfied that the CYP450 and MAPR knockout phenotypes are due to a primary role in steroidogenesis. The reviewer recommends creating a StAR knockout parasite. While I believe that creation of the StAR knockout parasite line is beyond the scope of this current work, the additional functions of CYP450 and MAPR in other systems need to be clearly described. The manuscript should contain an explicit statement about the limitations of the study, and the possibility that the phenotypes could be due to alternate functions. While the following statement is in the rebuttal letter “At this stage, we agree that we cannot rule out the possibility that the essentiality ofTgCYP450mt is due to its potential involvement in various other biosynthetic pathways, and not solely in steroidogenesis. We made this statement clearer in the revised version.” This statement could not be found in the main text of the manuscript.

The editorial changes recommended by reviewers 1 and 2 should be implemented to improve the clarity of the paper.

Lastly, authors should state their rationale for leaving Red-Green pseudo-coloring for the microscopy images, given the frequency of colorblindness in the population or change the pseudocoloring for IFA images.

Sincerely,

Aoife T. Heaslip, Ph.D

Guest Editor

PLOS Pathogens

Margaret Phillips

Section Editor

PLOS Pathogens

Kasturi Haldar

Editor-in-Chief

PLOS Pathogens

orcid.org/0000-0001-5065-158X

Michael Malim

Editor-in-Chief

PLOS Pathogens

orcid.org/0000-0002-7699-2064

Reviewer Comments (if any, and for reference):

Reviewer's Responses to Questions

**Part I - Summary**

Reviewer #2: The authors have addressed my queries and comments on the first version of the manuscript. I have only a few minor comments for their consideration, none of which require further experimentation. I congratulate them on completing an interesting study.

Reviewer #3: All major comments from original review remains.

Reviewer #4: In this manuscript, Asadi and colleagues describe the role of two proteins found in the parasitic protozoan Toxoplasma gondii, TgCYP450mt and TgMAPR. The former has homology to membrane-bound type I cytochrome P450 oxidases (CYP450) and the latter to members the membrane-associated progesterone receptor (MAPR) family. In mammals, synthesis of cholesterol-derived metabolites can occur via CYP450 proteins, which interact with MAPRs.

The authors detect hormone steroids in Toxoplasma, and using BLAST they were able homologs to find a single CYP450 homolog, and a single MAPR homolog. Both proteins localize to the parasite’s mitochondrion, and the authors show that the proteins interact with each other by proximity ligation assay. Generating a clean TgCYP450mt knock out was not possible, therefore the authors generate a conditional mutant instead. In the absence of TgCYP450mt, the parasites viability is compromised in vitro and their virulence in mice is attenuated.

Then the authors focus their efforts in TgMAPR. They generate an anti-TgMAPR antibody and use it for their phenotypical analyses. They also generate a recombinant TgMAPR and show the protein dimerizes and it is able to bind heme in vitro. They do an elegant experiment showing that TgMAPR can restore the growth of yeast lacking Dap1, an enzyme involved in ergosterol synthesis. A clean TgMAPR knock out strain was generated, and the authors show that ΔTgMAPR parasites grow slowly, and have invasion, egress and cell cycle defects. They also have an attenuated virulence. In the absence of TgMAPR, the parasites present aberrant morphological defects, with misplaced organelles and defects in cell division. Interestingly, after 7 weeks in culture, the ΔTgMAPR strain seems to lose most of its defects with the exception of mice virulence. The ΔTgMAPR and ΔTgMAPR strain after 7 weeks in culture have both attenuated virulence. Finally the authors show that both the conditional TgCYP450mt and ΔTgMAPR strains have reduced levels of steroids when compared to wild type parasites.

This is a very big manuscript, with a lot of interesting data pointing towards the ability of a single-celled parasite to metabolize cholesterol. The findings could have implications from an evolutionary and a therapeutic perspective. However, the manuscript feels disorganized. I feel that there is a lot of information (especially in the introduction) that is not relevant and just make everything harder to understand. Most of the figures and supplementary figures have not been harmonized, as there are different color palettes, styles and fonts. For example, the graph in figure 2C has a completely different style from the rest of the figure.

Some findings that should be included in the main text are included in the supplementary and vice versa. There are also several scale bars missing. Finally, the manuscript has what I think are two main parts: one about TgCYP450mt and the other one about TgMAPR. Compared to TgMAPR, the authors barely studied TgCYP450mt, as they generated a conditional knock down strain and only did general fitness assays. Why is that? Have the authors considered to look at mitochondrial morphology upon TgCYP450mt knock down? I think that will help to balance the manuscript The authors could alternatively reorganize the text and start first with the “big” part of the project, TgMAPR, and finish off with the part that feels more preliminary which involves TgCYP450mt (or maybe the authors are working on TgCYP450mt?).

**Part II – Major Issues: Key Experiments Required for Acceptance**

Reviewer #2: None

Reviewer #3: Many of the original concerns/points remain:

Major points:

Role of steroidogenesis (SG) in Toxoplasma gondii:

Using TgMAPR KO and iKDs of TgCYP450mt that are deficient for growth and have decreased levels of several of the steroids measured, the authors argue that steroidogenesis is essential for Toxoplasma growth and proliferation. However, steroidogenesis is not required for the survival of a eukaryotic cell (most cells in the human body are for example non-steroidogenic); and it is possible that these proteins have functions independently of SG that contributes to the negative effects of their deletion.

1) TgMAPR is a receptor that in mammals is not essential for steroidogenesis — why did the authors investigate this protein and not STAR that is required to transport cholesterol into the mitochodnria that is required for SG?

The answer the authors provide is not sufficient to address this point. As it stands, the manuscript supports that steroidogenesis occurs in Toxoplasma, but does not provide evidence that it plays an important role. Steroidogenesis has been shown to be essential at the level of the organismal. Toxoplasma is not a multicellular organism. Rather, the authors show that MAPR and CYP450 are important in Toxoplasma. As the authors’ comment: “Unlike mammalian cells, TgCYP450 is the only CYP450 enzyme present in Toxoplasma, and is essential in vitro and to survive in mice” Given that CYP450 enzymes act on several substrate, that there is only one in Toxoplasma would suggest that CYP450 in Toxoplasma has functions beyond steroidogenesis. The ablation of STAR in Toxoplasma would have been one strategy to address this.

Furthermore, the paper’s introduction of steroidogenesis still does not mention StAR and thus provides an unusually selective view on this topic. Otherwise please provide a more detailed rebuttal to this point. Were the role of StAR introduced earlier, the identification of a StAR domain containing Toxoplasma gene by sequence homology could be included in the results section (now only mentioned in discussion) which could be used to argue in support of the proposed model (that steroidogenesis is important for Toxoplasma tachyzoites).

2) Have the authors supplemented cells infected with SG-mutants (TgMAPR KO and iKDs of TgCYP450mt) with the steroids for which they are deficient? Does this rescue growth and or development?

The authors comment: “We believe this pathway is of central importance as the growth defects of

ΔTgMAPR and iΔTgCYP450mt cannot be rescued by culture medium with serum containing steroids”

Experimental data showing this needs to be included in the manuscript with a rationale how or why, if steroid production was an essential function for toxoplasma, its viability is not rescued by exogenous steroids. A good experimental design would be specific supplementation with the steroids found to be most abundant in Toxoplasma or those most affected by MAPR/CYP450mt ko, instead of response to presence or absence of serum.

3) Have the authors expressed heme-binding mutants and or enzymatic mutants of TgMAPR KO and iKDs of TgCYP450mt, respectively? How does this affect growth?

It is regrettable these experiments were not added, as they would enhance the publication, but they are not essential.

4) The authors report an ‘adapted TgMAPR KO. Does this parasite line have higher levels of the steroids deficient in the TgMAPR KO? This would be the expected result if steroids are essential for parasite proliferation.

Not addressing this point (similar to 1 + 2) by performing the suggested experiment remains a major weakness of the manuscript; this could be one way to address the importance of steroids (ie. if the adapted TgMAPRKO is still able to make steroids; however if it too is steroid sufficient that would argue against an important role for steroids in parasite proliferation).

5) TgMAPRKO parasites appear to form cyst-like structures that are positive for TRITC-lectin (fig. 5B). The authors conclude that these are not cysts because they are BAG1-deficient but have other markers of cyst formation been examined? Has cyst formation in surviving mice been examined? Have the authors examined cyst-formation in Type II MAPR KOs?

Addressed.

6) Effect of TgMAPR and TgCYP450mt on steroidogenesis:

The authors show that parasites deficient for TgMAPR and TgCYP450mt have decreased levels of various steroids (Fig. 11). How was this experiment conducted? Given the dramatically different growth rates, how can the authors be sure this is reflecting a difference in steroid synthesis rather than development-stage synthesis (i.e. more SG occurs at later stages in the lytic cycle). Furthermore, how can the authors be sure that their deficiency is directly affecting SG rather than indirectly affecting mitochondria? Can enzymes more downstream in SG be ablated? Can aromatase inhibitors be tested for their effects on Toxoplasma SG and growth?

Given the small scale of the observed effects, to convincingly demonstrate steroidogenesis impairment by the knockouts the experiment requires more replicates to allow for statistical analysis (Fig. 9). In principle, a non-targeting ko control is required for this kind of experiment and the omission of such warrants a justification. Furthermore, it is unclear why the most abundant steroid detected from toxoplasma, 17-hydroxypregnenolone, was not used in the wt vs ko comparison. The authors are strongly urged to improve this data as it represents a cornerstone of their findings.

All minor comments were addressed.

Reviewer #4: N/A

**Part III – Minor Issues: Editorial and Data Presentation Modifications**

Reviewer #2: I picked up a few minor typographical errors and have some other queries that could be addressed to enhance clarity of the manuscript (line numbers correspond to those of the marked-up copy):

1. Line 59. “severe growth [impairment] in cultured cells”

2. Line 75. “viable [in] vitro”

3. Line 144. “www.tox[o]db.org”. Also check line 181: .org not .com

4. Line 196. “TgCYP450 shares high sequence identity with mitochondrial CYP450 homologs present in fellow members of the Sarcocystidae family” What is the basis for concluding the CYP450 homologs in other Sarcocystidae are mitochondrial? Is there experimental evidence for this? If so, should cite. If not, then reword.

5. Line 229 and Figure 1E. “Data showed that all mice infected with iΔTgCYP450 parasites pretreated with ATc remained alive at Day 22”. Is this correct? From the graph, ~ 80% of mice in the +ATc condition remain alive.

6. Figure 1E and Line 231. It is unclear from the description in the results and methods exactly how these experiments were performed. Did the authors pre-treat the parasites with ATc (which is what the text seems to suggest)? Did they then maintain mice with ATc in the drinking water? This should be clarified in the methods. If they pre-treated mice with ATc, then they cannot conclude that “TgCYP450mt is essential for parasite … survival in mice”. Rather, this experiment is testing the viability of parasites following pre-treatment in ATc.

7. Line 243. “instead exerting their function at a nongenomic level”. The meaning of “nongenomic level” isn’t clear.

8. Lines 260-265. The alignment of the TgMAPR protein to the human homolog in the C-terminal region is not well-supported (eg the Y180 and RK192-193 residues in TgMAPR do not align with the proposed equivalent residues in the human protein). Concluding that the phosphorylation sites are conserved (especially in the absence of evidence that these are phospohorylation sites) is not well supported by the alignment.

9. Figure 2B. Should include molecular mass markers on these western blots.

10. Line 397. “the mitochondrion exhibiting large cristae (Fig. 4A, panel iv). The mitochondria in this image are quite small and the cristae don’t seem obvious. Can the authors supply an image with better resolution and/or an image with greater magnification?

11. Line 471. Do the authors mean to refer to Fig. 4A here?

12. Line 597. “this suggests a contribution of TgMAPR to parasite adaptation to a checkpoint that controls G1 progression”. The meaning here isn’t clear. Progression into or through G1 phase? It is conceivable that parasites are defective in an element of mitosis or, as the earlier part of this sentence and some of the authors’ data suggest, daughter cell formation/cytokinesis.

Reviewer #3: All previous minor issues addressed.

Reviewer #4: Here are my comments, in order of appearance and not in order of importance:

1) Out of pure curiosity, how come the authors do not detect any sterols in the host cells? Aren’t they supposed to also contain sterols?

2) In the introduction, nothing is mentioned about cholesterol-derived metabolites in protozoans. It is because not a lit is know? The authors could mention that and it will help with the novelty of their findings.

3) Line 132: why is it of interest to know that PGRMC1 is overexpressed in malignant cancer cells and localizes to the mitochondria? The authors could either remove this sentence or explain why does this have a link with Toxoplasma gondii.

4) Line 144: How many genes in Toxoplasma contain ligand-binding domains typifying nuclear hormone receptors or the sequence motifs (e.g., Zn-finger, C4-type steroid receptor? I think the authors should be more specific. This will contribute to the evolutionary aspect of this manuscript, as we would expect protozoan to have a reduced number of receptors compared to mammals.

5) Line 187: Sequence similarity searches are limited, and structural homology should also be implemented: the authors should use HHPRED (https://toolkit.tuebingen.mpg.de/tools/hhpred) to check for structural homology. In fact, can see with HHPRED that the TgCYP450mt shares structural homology to cytochrome p450 4F22 but not to the other proteins mentioned in the text.

6) Line 223: I think the WB showing depletion of TgCYP450mt should be included in the main text, in figure 1. The authors could remove some of the panels of figure 1B and put them in the supplementary figures instead (since they all show colocalization of the protein with TOM40).

7) Line 252: How was the predicted transmembrane domain information obtained? The Toxoplasma database does not predict any TM.

8) Line 280: I would add that the antibody was generated in mice.

9) Line 304: The authors use the hyperLOPIT data to help them with the characterization of TgCYP450mt (line 185), but then do not do that with TgMAPR They should include it. It seems that the predicted localization of TgMAPR is mitochondrion – soluble, but the authors show otherwise.

10) Line 312: I do not understand why the authors generated an TgMAPR-HA to confirm the localization of the protein. The localization of TgMAPR with the anti-TgMAPR antibody already shows that? Or I am missing something. To me, the only experiment that the authors should perform is an IFA with their antibody and anti-HSP70.

11) Line 331, comments for figure S6A: This is a very important and exciting experiment. However I would like to see a lane where the authors just use the parental yeast strain, and a loading control I think that is the bare minumus for a western blot. Additionally, did the authors try to localize the protein by IFA? Since DAP1 localizes to the ER, with would be interesting to see where TgMAPR localizes. Would it be going to the ER or to the mitochondria?

12) Line 340, pathway in Figure S6B: The pathway is unnecessarily complicated. It is also small, clogged and hard to read. Please highlight the enzyme, and just add the precursor and final products along with the intermediates you mention in the text.

13) Line 373: the authors should be more precise when they mention “poor fitness of ΔTgMAPR parasites”. Since they did the experiments, I think they are refering to invasion and egress defects.

14) Line 555: a quick search in the Toxoplasma database shows that there are three annotated StAR domain-containing proteins: tgme49_223150, 231000 and 236660. What made the authors choose the one they mention over the other two?

PLOS authors have the option to publish the peer review history of their article (what does this mean?). If published, this will include your full peer review and any attached files.

Reviewer #2: No

Reviewer #3: No

Reviewer #4: No

Figure Files:

Data Requirements:

Reproducibility:

References:

---

## [Editor Report · Decision Letter 2]

19 Jul 2023

Dear Dr. Coppens,

We are pleased to inform you that your manuscript 'Function and regulation of a steroidogenic CYP450 enzyme in the mitochondrion of Toxoplasma gondii' has been provisionally accepted for publication in PLOS Pathogens.

Best regards,

Aoife T. Heaslip, Ph.D

Guest Editor

PLOS Pathogens

Margaret Phillips

Section Editor

PLOS Pathogens

Kasturi Haldar

Editor-in-Chief

PLOS Pathogens

orcid.org/0000-0001-5065-158X

Michael Malim

Editor-in-Chief

PLOS Pathogens

orcid.org/0000-0002-7699-2064
---

## [Editor Report · Acceptance letter]

23 Aug 2023

Dear Dr. Coppens,

We are delighted to inform you that your manuscript, "Function and regulation of a steroidogenic CYP450 enzyme in the mitochondrion of Toxoplasma gondii," has been formally accepted for publication in PLOS Pathogens.

Best regards,

Kasturi Haldar

Editor-in-Chief

PLOS Pathogens

orcid.org/0000-0001-5065-158X

Michael Malim

Editor-in-Chief

PLOS Pathogens

orcid.org/0000-0002-7699-2064